# Type I interferon regulates proteolysis by macrophages to prevent immunopathology following viral infection

Amanda J. Lee[1,ʘ,‡], Emily Feng[1,ʘ,‡], Marianne V. Chew[1], Elizabeth Balint[1], Sophie M. Poznanski[1], Elizabeth Giles[1], Ali Zhang[2,3], Art Marzok[2,3], Spencer D. Revill[1,4], Fatemeh Vahedi[1], Anisha Dubey[1,4], Ehab Ayaub[1,3,4], Rodrigo Jimenez-Saiz[1], Joshua J. C. McGrath[1], Tyrah M. Ritchie[1], Manel Jordana[1], Danny D. Jonigk[5], Maximilian Ackermann[6,7], Kjetil Ask[1,4], Matthew Miller[2,3], Carl D. Richards[1], Ali A. Ashkar[1,‡*]

1 Department of Medicine, McMaster Immunology Research Centre, McMaster University, Hamilton, Ontario, Canada, 2 Department of Biochemistry and Biomedical Sciences, McMaster University, Hamilton, Ontario, Canada, 3 Michael G. DeGroote Institute for Infectious Disease Research, McMaster University, Hamilton, Ontario, Canada, 4 Department of Medicine, Firestone Institute of Respiratory Health, McMaster University, Hamilton, Ontario, Canada, 5 Institute of Pathology, Hannover Medical School, Member of the German Center for Lung Research (DZL), Biomedical Research in Endstage and Obstructive Lung Disease Hannover (BREATH), Hannover, Germany, 6 Institute of Pathology and Molecular Pathology, Helios University Clinic Wuppertal, University of Witten/Herdecke, Wuppertal, Germany, 7 Institute of Functional and Clinical Anatomy, University Medical Center of the Johannes Gutenberg-University Mainz, Mainz, Germany

ʘ These authors contributed equally to this work.
‡ AJL and EF are co-first authorship on this work. AAA is Lead contact on this work.
* ashkara@mcmaster.ca

**Data Availability Statement:** All relevant data are within the paper and its Supporting Information files.

## Abstract

The ability to treat severe viral infections is limited by our understanding of the mechanisms behind virus-induced immunopathology. While the role of type I interferons (IFNs) in early control of viral replication is clear, less is known about how IFNs can regulate the development of immunopathology and affect disease outcomes. Here, we report that absence of type I IFN receptor (IFNAR) is associated with extensive immunopathology following mucosal viral infection. This pathology occurred independent of viral load or type II immunity but required the presence of macrophages and IL-6. The depletion of macrophages and inhibition of IL-6 signaling significantly abrogated immunopathology. Tissue destruction was mediated by macrophage-derived matrix metalloproteinases (MMPs), as MMP inhibition by doxycycline and Ro 28–2653 reduced the severity of tissue pathology. Analysis of post-mortem COVID-19 patient lungs also displayed significant upregulation of the expression of MMPs and accumulation of macrophages. Overall, we demonstrate that IFNs inhibit macrophage-mediated MMP production to prevent virus-induced immunopathology and uncover MMPs as a therapeutic target towards viral infections.

**Funding:** This work was funded by the Canadian Institutes of Health Research (CIHR) (20008285 to A.A.A). A.A.A holds a Tier 1 Canadian Research Chair in Natural Immunity and NK Cell Function. A. J.L and S.M.P are recipients of a CIHR Vanier Canada Graduate Scholarship. E.F and E.B, are supported by a Master's Canadian Graduate Scholarship and Master's Ontario Graduate Scholarship. The funders had no role in study design, data collection and analysis, decision to publish, or preparation of the manuscript.

**Competing interests:** The authors have declared that no competing interests exist.

## Author summary

Dysregulated immune responses and their associated pathologies are the culprit of severe disease symptoms in response to viral infections. The ability to properly regulate effective and controlled immune responses is a critical feature of preventing severe disease outcomes. Type I interferons are antiviral signaling molecules known to induce potent antiviral immune responses; however, their ability to suppress pathogenic immune responses is poorly understood. Employing a vaginal HSV-2 infection model in mice, we show that type I IFN signaling is critical to preventing the development of severe tissue pathology by suppressing the pathogenic functions of macrophages. In the absence of type I IFNs, these unleashed macrophages produce MMPs that can degrade tissue structure. We show that inhibiting MMPs reduces the severity of immunopathology. We further provide evidence that influenza infection in mice, as well as severe COVID-19 infection in humans, is linked to macrophage and MMP-mediated tissue destruction. Together, our study describes a distinct mechanism through which type I IFNs regulate pathogenic immune responses, and defines MMPs as a potential therapeutic target during severe viral infections.

## Introduction

Effective early innate immune responses are crucial in determining the clinical outcomes of viral infections. Dysregulated inflammatory immune responses will not only impede early viral replication, but ultimately lead to immune-mediated tissue pathology. Type I interferons (IFNs) are master regulators of the early innate immune response and determinants of disease outcome against viral infection [1,2]. For instance, during genital herpes simplex virus type 2 (HSV-2) infections, type I IFN induction stimulates early antiviral immunity through promoting monocyte recruitment and Natural Killer (NK) cell production of IFN-γ, necessary for host defense [3]. The absence of type I IFN signaling results in increased viral replication and reduced survival to vaginal HSV-2 infection [3,4].

Studies of highly pathogenic viral infections, including severe acute respiratory syndrome coronavirus type 2 (SARS-CoV)-2 and Ebola virus (EBOV), demonstrate that early type I IFN induction consistently correlates with disease tolerance [5–7]. These immunopathogenic responses often occur independently of viral load and are the result of dysregulated heightened inflammation [8]. Type I IFNs have demonstrated an ability to control potentially pathogenic immune responses. During HSV-2 infection, type I IFNs both induce IFN-γ production by NK cells, but also inhibit their production during later stages of infection [9]. In a mouse model of influenza infection, type I IFNs have also been demonstrated to inhibit various sources of immune-mediated pathology [10,11]. Understanding how type I IFNs can both promote viral clearance, but also control inflammation to improve disease outcomes, is critical to understanding the mechanisms of disease pathology and develop effective treatments for viral infections.

The immune-mediated tissue pathology during viral infection is fueled by pro-inflammatory cytokines such as IL-6, TNF-α and IL-1β [12]. These responses have been recognised in not just infectious diseases, but also in graft-versus-host disease and various chronic inflammatory disorders [13–15]. Of these cytokines, IL-6 signaling has been particularly emphasized for its ability to regulate various aspects of both protective and pathogenic immune responses. Elevated IL-6 has been specifically highlighted as a contributor to pathogenesis during severe COVID-19 and IAV, and many efforts to block IL-6 signaling to improve disease outcomes are underway [16–19]. Inhibiting IL-6 responses can impair leukocyte recruitment and

dampen T cell and neutrophil antiviral responses [20–24]. Furthermore, IL-6 has also been described as the driving force behind CRS [25]. However, despite the well-accepted relationship between IL-6 and inflammation, the specific role of IL-6 in the development of immune-mediated tissue pathology is poorly defined. We hypothesize that type I IFNs play a central role in suppressing cytokine-driven hyperinflammation during infection.

Here, we report that in the absence of IFNAR, mucosal viral infection leads to the development of significant immune-mediated tissue pathology in mice. We found that type I IFN signaling suppressed IL-6 production. In the absence of IFN-mediated suppression, IL-6 was critical in inciting immunopathology through the induction of phagocytic macrophages and their production of matrix metalloproteinases (MMPs). Interestingly, direct inhibition of MMP proteolytic activity was sufficient to limit immunopathology during infection. Further, we show that the immunoregulatory function of type I IFNs occurs independent of the location and type of viral infection. Overall, we define a previously unidentified immunomodulatory function of type I IFN, which acts as a suppressor of IL-6-induced macrophage and MMP-mediated immunopathology to prevent tissue damage and increase disease tolerance. We further identify MMPs as a promising therapeutic target in limiting damage and promoting survival following mucosal viral infections.

## Results

### Absence of type I IFN signaling is associated with severe genital tissue destruction during HSV-2 infection

The antiviral function of type I interferons is well known; however, less is known about their immune regulatory functions during viral infections. In response to CpG intravaginal stimulation, WT mice show early IFN-β production at 6 and 12 hrs [26], and we observe a peak production of IFN-α 48 hrs post HSV-2 infection (S1A Fig). To investigate the role of type I IFN signaling in the regulation of immune response to viral infection, we infected C57BL/6 *Ifnar*[-/-] mice intravaginally with HSV-2 (Fig 1A). *Ifnar*[-/-] mice displayed considerable and observable swelling of the vaginal tract at 3 days post-infection (dpi) compared to WT mice following HSV-2 infection (Fig 1B). Closer histomorphological analysis of vaginal tissue cross-sections revealed extensive destruction of mucosal and sub-mucosal structures including the loss of epithelial lining, haemorrhaging, and break down of vaginal tissue integrity in *Ifnar*[-/-] mice (Fig 1C and 1E). At baseline, we found no inflammation in either the WT or *Ifnar*[-/-] vaginal mucosa cross-sections. Picosirius red (PSR) staining demonstrated a significant decrease in collagen in *Ifnar*[-/-] mice at 3 dpi compared to WT mice and to baseline, indicative of collagen degradation (Fig 1D and 1F). To confirm these findings were not due to inherent long-term IFNAR deficiency, we administered an α-IFNAR neutralizing antibody, and observed a similar level of vaginal pathology to *Ifnar*[-/-] mice (Fig 1G and 1H). Characterization of inflammation in *Ifnar*[-/-] mice revealed a significant increase in the total number of CD45+ cells in the vaginal tissue compared to WT mice at 3 dpi, coinciding with the observed pathology (Fig 1I). These results suggest that type I IFN signaling suppresses immune-mediated pathology in the vaginal mucosa.

Loss of IFNAR can result in failure to control viral replication [3,27,28]. To determine if an increase in viral replication was responsible for mucosal tissue pathology observed in *Ifnar*[-/-] mice, we compared HSV-2 infection in *Ifnar*[-/-] and *Il15*[-/-] mice. Loss of IL-15 during viral infection impairs NK cell maturation and innate IFN-γ production, resulting in increased HSV-2 viral titers compared to WT mice [28]. In agreement with previous results [28], viral titers were both increased in *Il15*[-/-] and *Ifnar*[-/-] mice compared to WT mice (Fig 1L). However, unlike *Ifnar*[-/-] mice, analysis of *Il15*[-/-] vaginal tissue cross-sections with intact type I IFN signaling revealed little evidence of tissue destruction and inflammation similar to WT mice (Fig 1J

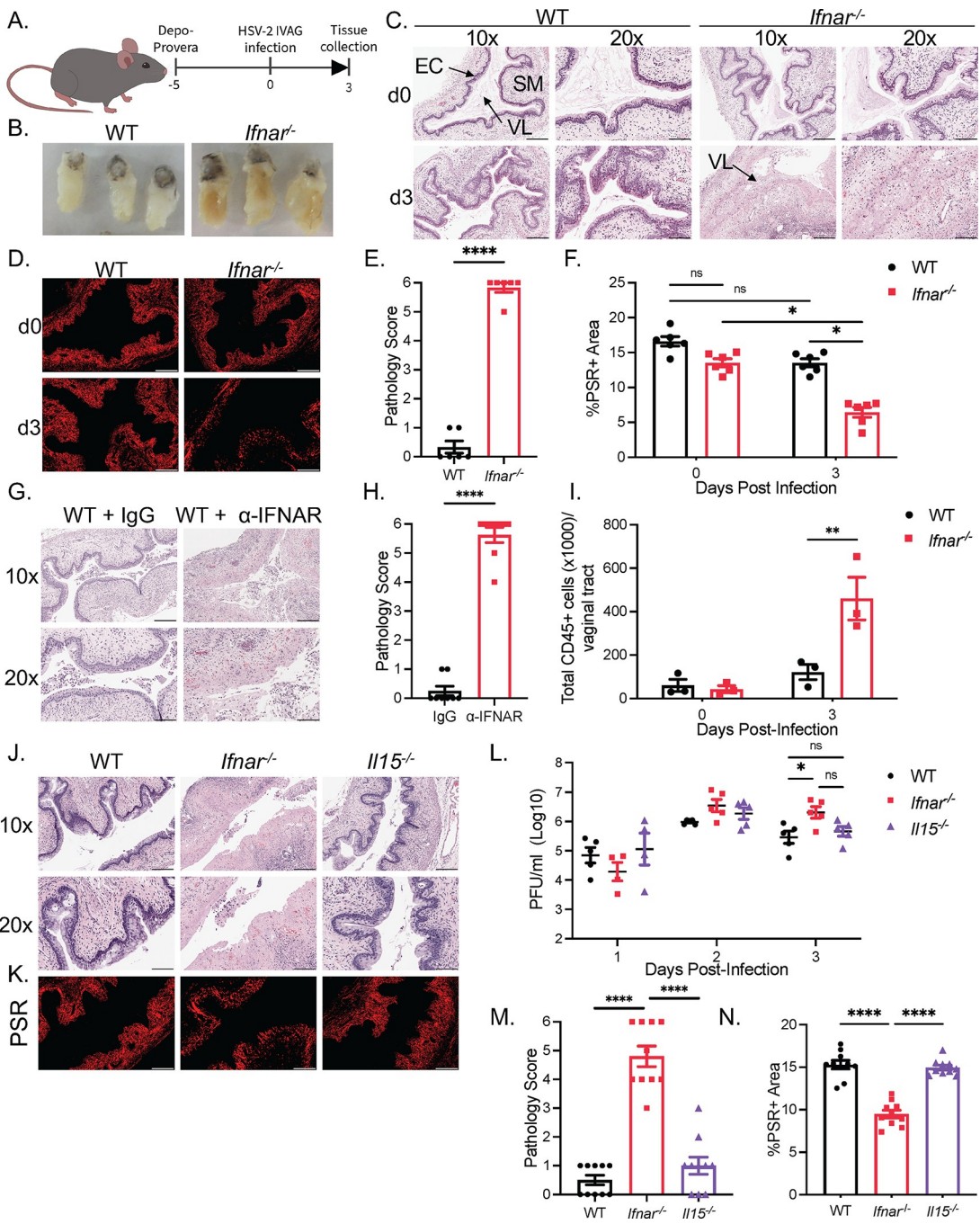

**Fig 1. Absence of type I IFN signaling is associated with severe genital tissue destruction during HSV-2 infection.** (A) Schematic of WT and *Ifnar*[-/-] mice infected with 10[4] PFU of HSV-2 333 intravaginally. (B) Whole vaginal tissue of WT and *Ifnar*[-/-] mice 3 dpi. (C) Histological analysis of WT and *Ifnar*[-/-] mice vaginal cross-sections with H&E at 0 and 3 dpi (n = 5). (D) PSR staining of WT and *Ifnar*[-/-] mice vaginal cross-sections at 0 and 3 dpi. (E) Quantification of pathology of (C) at 3 dpi. (F) Quantification of %PSR+ area to total vaginal tissue area quantification in (D). (G and H) H&E staining (G) and pathology score (H) of *Ifnar*[-/-] mice given isotype control or α-IFNAR vaginal cross-sections at 0 and 3 dpi (n = 5). (I) Quantification of total CD45+ cells in vaginal tissue through flow cytometry at 0 and 3 dpi (n = 3). (J) Comparison of WT, *Ifnar*[-/-] and *Il15*[-/-] mice vaginal histology at 3 dpi with H&E staining (n = 5). (K) PSR staining of WT, *Ifnar*[-/-] and *Il15*[-/-] mice PSR staining at 3 dpi. (L) Quantification of viral titers in vaginal washes WT, *Ifnar*[-/-] and *Il15*[-/-] mice. (M) Pathology score of (J). (N) Quantification of % PSR+ area to total vaginal tissue area quantification in (K). 10x scale bar represents 200 μm, 20x and PSR scale bar represents 100 μm. Data in (E), (F), (H), (I), (L), (M), (N) are represented as mean ± SEM. $^*p < 0.05$, $^{**}p < 0.01$, and $^{****}p < 0.0001$. (F, I, L two-way ANOVA; M, N, one-way ANOVA; E, H, two-tailed t-test) See also S1 Fig.

and 1M). Furthermore, there was no difference in PSR staining between WT and *Il15*-/- vaginal tissue cross-sections at 3 dpi, and both showed significantly higher levels of collagen in comparison to *Ifnar*-/- mice (Fig 1K and 1N). These results suggest that the elevated viral load was not responsible for the increased tissue pathology shown in *Ifnar*-/- mice. They further demonstrate that type I IFNs possess a critical function separate from their antiviral properties in preventing pathology in the vaginal mucosa.

Since IFNAR deficiency is also associated with a deficiency in NK cell function and innate IFN-γ production [28], we examined if restoring type II IFNs can abrogate the HSV-2-induced tissue pathology. As expected, *Ifnar*-/- mice showed no production of IFN-γ during the first 3 days of infection (S1B Fig). Administration of IL-12 and IL-18 induced IFN-γ production at 2 and 3 dpi in *Ifnar*-/- mice, which rescued vaginal immunopathology and collagen staining, and reduced viral titers (S1C–S1F Fig). This effect was dependent on NK cells, as depletion with α-NK1.1 mAb reversed the protection afforded by IL-12 and IL-18 administration in *Ifnar*-/- mice (S1C Fig). This suggests that the induction of type II IFNs is sufficient to prevent the development of viral-induced pathology in the absence of type I IFN signaling. Meanwhile, *Il15*-/- mice, that possess no early IFN-γ production, but intact type I IFN signaling, show no development of tissue pathology. These results suggest that innate IFN-γ signaling and type I IFN signaling possess redundant roles in suppressing virus-induced pathology, as presence of either type I or type II IFN signaling is sufficient to prevent immunopathology.

## Vaginal immunopathology in *Ifnar*-/- mice is independent of ILC2s or TH2 CD4+ T cells

It has been shown that deficiency or absence of type I IFN signaling is associated with an increased inflammatory response, and reduced type I IFN induction is correlated with disease severity in COVID-19 patients [7,29]. However, how type I IFN signaling prevents excessive inflammation and tissue damage during viral infections is less clear. Some have speculated that type I IFNs can suppress inflammatory responses mediated by TH2 cells and group 2 innate lymphoid cells (ILC2s) in response to allergens or infections [11,30]. To assess the involvement of any lymphoid-derived cells in vaginal immunopathology, we employed NOD-*Rag2*-/-*IL2rγ*-/- (NRG) mice lacking any lymphoid lineage cells, including ILC2s and CD4+ T-cells. Blocking IFNAR signaling through administering an α-IFNAR antibody in NRG mice, like C57BL/6 *Ifnar*-/- mice, was sufficient to induce significant vaginal immunopathology and loss of collagen staining, compared to isotype-matched Ig controls (Fig 2A–2C). To confirm these findings in *Ifnar*-/- mice, we detected only a modest increase in proportion of ILC2 cells between WT and *Ifnar*-/- mice at 2 dpi, and no difference in the total number of ILC2 cells between the two groups (Figs 2D, 2E and S2A). We also detected no differences in the levels of vaginal IL-33, a potent stimulator of ILC2 cells, between WT and *Ifnar*-/- mice 1 dpi (S2B–S2D Fig). WT and *Ifnar*-/- mice also had comparable levels of CD3+ T cells, both in proportion and total number, in their vaginal mucosa at 2 and 3 dpi (Figs 2F, 2G and S2E). Furthermore, depletion of CD4+ T cells through administering an α-CD4 depleting antibody in *Ifnar*-/- mice resulted in no observable difference in the degree of immunopathology compared to control mice (S2F–S2J Fig). These experiments collectively indicate that virus-induced vaginal immunopathology in the absence of IFNAR occurs independently of lymphoid lineage cells, including both ILC2s and CD4+ T-cells, as well as NK cells and CD8+ T cells.

## Phagocytic cells are required for HSV-2-induced tissue immunopathology in *Ifnar*-/- mice

Since we observed that neither ILCs nor a lymphoid cell-mediated type II immune response were responsible for vaginal pathology in the absence of type I IFN signaling, we addressed

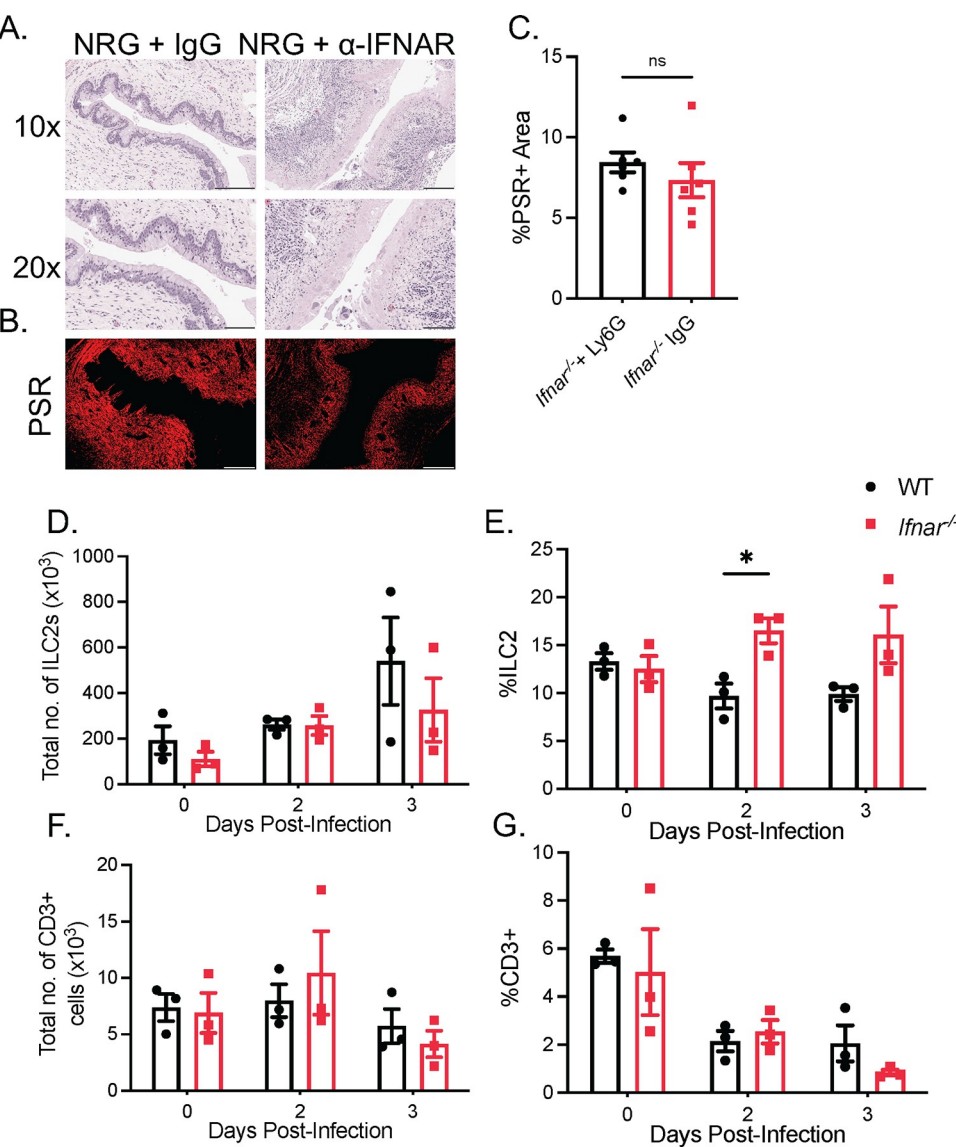

**Fig 2. Vaginal immunopathology in *Ifnar*$^{-/-}$ mice is independent of ILC2s or TH2 CD4$^+$ T cells.** (A) H&E staining of vaginal cross-sections of NRG mice + α-IFNAR or isotype control 3 dpi (n = 5). (B and C) PSR staining (B) and quantification of PSR+ to total vaginal tissue (C) of NRG mice + α-IFNAR or isotype control 3 dpi (n = 5). (D and E) Total number of CD45+Lin-ST2+CD90.2+ ILC2s (D) and proportion of ILC2s to total CD45+ cells (E) in vaginal tissue of WT and *Ifnar*$^{-/-}$ mice following HSV-2 infection (n = 3). (F and G) Total number of CD3+ T cells (F) and proportion of CD3+T cells to total CD45+ cells (G) in vaginal tissue (n = 3). 10x scale bar represents 200 μm, 20x and PSR scale bar represents 100 μm. Data in (C) to (D), (E), (F), and (G), are represented as mean ± SEM. $^*$p < 0.05. (D-G, two-way ANOVA; C two-tailed t test) See also S2 Fig.

whether immunopathology was induced by innate immune cells of the myeloid lineage. We examined the influx of neutrophils and eosinophils following infection of the vaginal mucosa of WT and *Ifnar*$^{-/-}$ mice. We detected a significant increase in the proportion and total number of neutrophils in the vaginal mucosa at 2 dpi in *Ifnar*$^{-/-}$ mice compared to WT mice (S3A–S3C Fig). To determine whether the increase in neutrophils in *Ifnar*$^{-/-}$ mice was responsible for HSV-2-induced tissue pathology, we depleted neutrophils using the α-Ly6G mAb in *Ifnar*$^{-/-}$ mice. There was significant depletion of neutrophils at 3 dpi in the blood, spleen, and vaginal

tissue of WT mice (S3D–S3F Fig). However, α-Ly6G administration did not abrogate vaginal tissue destruction and collagen degradation in *Ifnar*$^{-/-}$ mice (S3G–S3l Fig). Thus, while neutrophil recruitment is increased in the absence of IFNAR, they are not required for vaginal tissue pathology. We also examined eosinophil populations in the vaginal mucosa and observed no difference in the proportion or total cell numbers between WT and *Ifnar*$^{-/-}$ mice, suggesting that eosinophils are also not involved in vaginal immunopathology (S3J–S3L Fig).

In instances of virus-induced lung immunopathology, macrophages and inflammatory monocytes (IM) have been implicated in the immunopathological process through the release of pro-inflammatory cytokines and the upregulation of TRAIL [31–33]. We have previously described that infiltration of IMs to HSV-2-infected vaginal tissue is impaired in *Ifnar*$^{-/-}$ mice (3). To determine if there were changes in macrophage populations, we examined levels of F4/80+ cells at 0 and 3 dpi in the vaginal tissue in WT and *Ifnar*$^{-/-}$ mice. Though we did not detect a difference in the proportion of macrophages, we found a significant increase in the total number of vaginal F4/80+ cells in *Ifnar*$^{-/-}$ mice at 3 dpi (Figs 3A, 3B and S3M). To investigate if macrophages contribute to tissue pathology, clodronate liposomes were administered to reduce the number of phagocytic macrophages (Figs 3C and S3N). Clodronate administration in *Ifnar*$^{-/-}$ mice markedly reduced tissue damage without impeding control of HSV-2 replication (Fig 3D–3F). To examine if depletion of macrophages could abrogate HSV-2 induced immunopathology in the absence of any lymphoid cells, we repeated the use of clodronate in NRG mice given α-IFNAR neutralizing Ab. This also resulted in significant abrogation of HSV-2-induced tissue damage (Fig 3G and 3H). These results demonstrate that type I IFNs regulate macrophage function to prevent or minimize tissue damage during HSV-2 infection.

## Type I IFNs regulate IL-6 production to prevent macrophage-mediated immunopathology during viral infection

To understand how type I IFNs regulate macrophages to prevent HSV-2-induced immunopathology, we measured the levels of various type II cytokines, proinflammatory cytokines, and chemokines in the vaginal washes following HSV-2 infection (Fig 4A). Loss of type I IFN signaling resulted in a very modest increase in the production of stereotypical type II cytokines IL-5 and IL-13 in *Ifnar*$^{-/-}$ mice at 3 dpi, and no change in IL-4 production (S4A–S4C Fig). At baseline, there was only a significant increase in IL-9 in uninfected *Ifnar*$^{-/-}$ mice (S4D Fig). On the other hand, we observed a substantial increase (6.54-fold) in the production of IL-6 at 3 dpi over WT mice (Fig 4B), coinciding with the pathology exhibited in *Ifnar*$^{-/-}$ mice. To confirm that IL-6 contributes to the HSV-2-induced tissue pathology, we examined whether neutralizing IL-6 in *Ifnar*$^{-/-}$ mice during HSV-2 infection could abrogate the tissue pathology (Fig 4C). Blocking IL-6 with a neutralizing α-IL-6 mAb led to significant abrogation of vaginal immunopathology, suggesting that IL-6 induces immunopathology in *Ifnar*$^{-/-}$ mice (Fig 4D and 4E). Similarly, *Il6*$^{-/-}$ mice administered α-IFNAR mAb also displayed decreased pathology compared to WT mice given α-IFNAR mAb (Fig 4G and 4H). Importantly, IL-6 induced pathology independently of viral replication, as viral titers did not change following α-IL-6 treatment (Fig 4F). We further investigated whether exogenous supplementation of recombinant IL-6 (rIL-6) was sufficient to induce pathology in WT mice (Fig 4I). Injection of rIL-6 in WT mice resulted in increased IL-6 levels in vaginal washes (Fig 4J). rIL-6 was not sufficient in inducing tissue destruction in WT mice in the absence of infection but induced low but significant level of epithelial destruction following HSV-2 infection (Fig 4K and 4L). Thus, we have shown that absence of type I IFN signaling during viral infection results in excessive IL-6 production, and that macrophages are crucial for the observed tissue pathology.

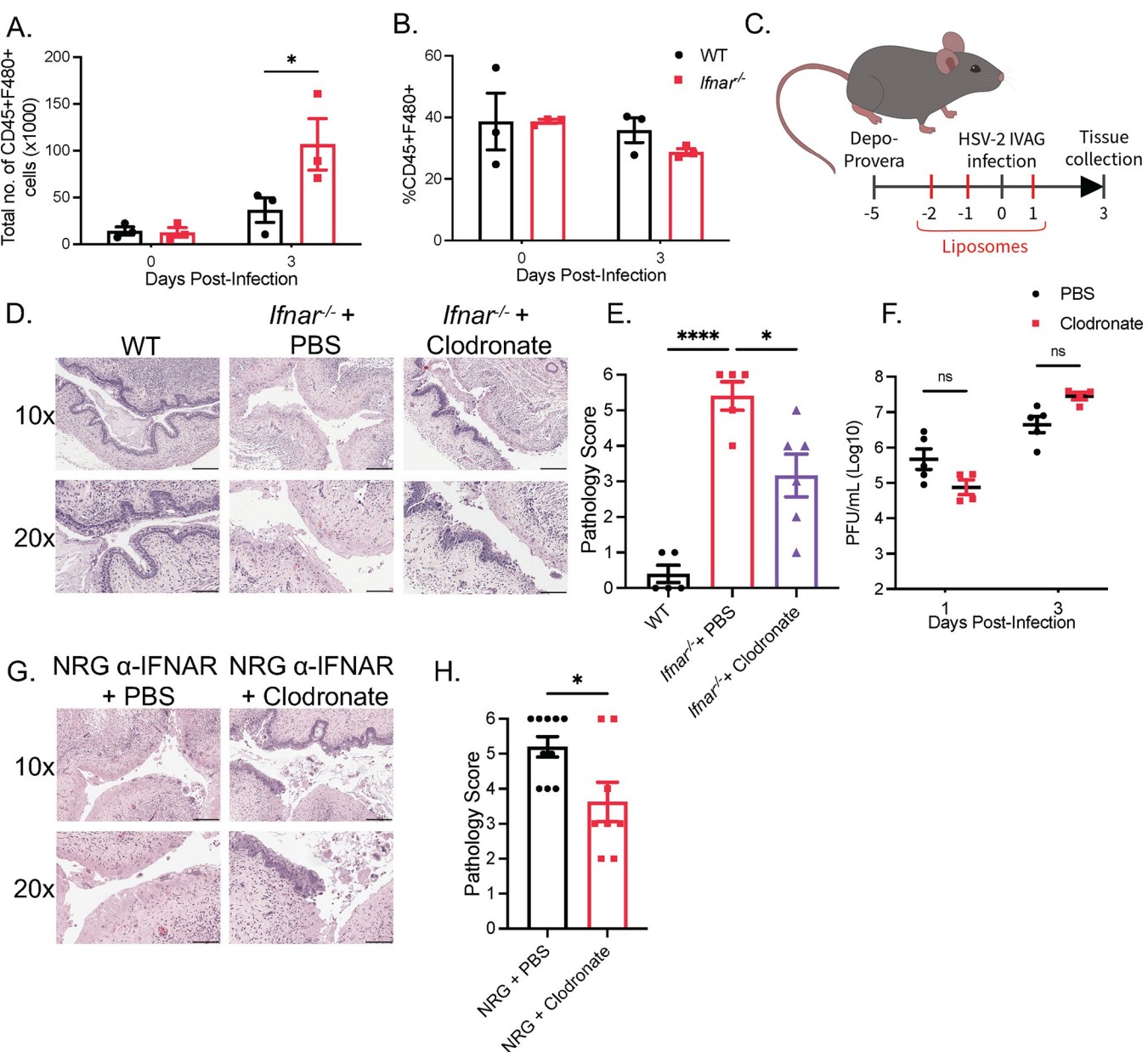

**Fig 3. Phagocytic cells are required for HSV-2-induced immunopathology in *Ifnar*<sup>-/-</sup> mice.** (A) Total number of CD45+F480+ cells in vaginal tissue following infection (n = 5). (B) %CD45+F480+ to total CD45+ cells in vaginal tissue of WT and *Ifnar*<sup>-/-</sup> mice at 0 and 3 dpi following HSV-2 infection (n = 5). (C) Schematic of *Ifnar*<sup>-/-</sup> mice and NRG mice receiving clodronate or PBS control liposomes. (D) H&E staining of vaginal cross-sections of HSV-2-infected WT and *Ifnar*<sup>-/-</sup> mice administered PBS or clodronate liposomes at 3 dpi (n = 5–6). (E) Pathological score of (D). (F) Viral titers of vaginal washes at 1 and 3 dpi of HSV-2-infected *Ifnar*<sup>-/-</sup> mice given PBS or clodronate liposomes (n = 4–5). (G) H&E staining of vaginal cross-sections of HSV-2-infected NRG + α-IFNAR Ab mice with PBS or clodronate liposomes at 3 dpi (n = 4–5). (H) Pathology score of (G). 10x scale bar represents 200 μm, 20x scale bar represents 100 μm. Data in (A), (B), (E), (F), (H) and are represented as mean ± SEM. *p < 0.05, and ****p < 0.0001 (A, B, F, two-way ANOVA; E, one-way ANOVA; H, two-tailed t-test). See also S3 Fig.

Elevated levels of IL-6 are often identified during severe viral infections. However, their exact role in regulating inflammation and pathology is uncertain. We investigated how IL-6 and macrophages caused the rapid and significant tissue damage following viral infection in *Ifnar*<sup>-/-</sup> mice. IL-6 signaling has the ability to potentiate macrophage function and polarisation, and promote leukocyte recruitment [22,34,35]. Depletion of macrophages through clodronate

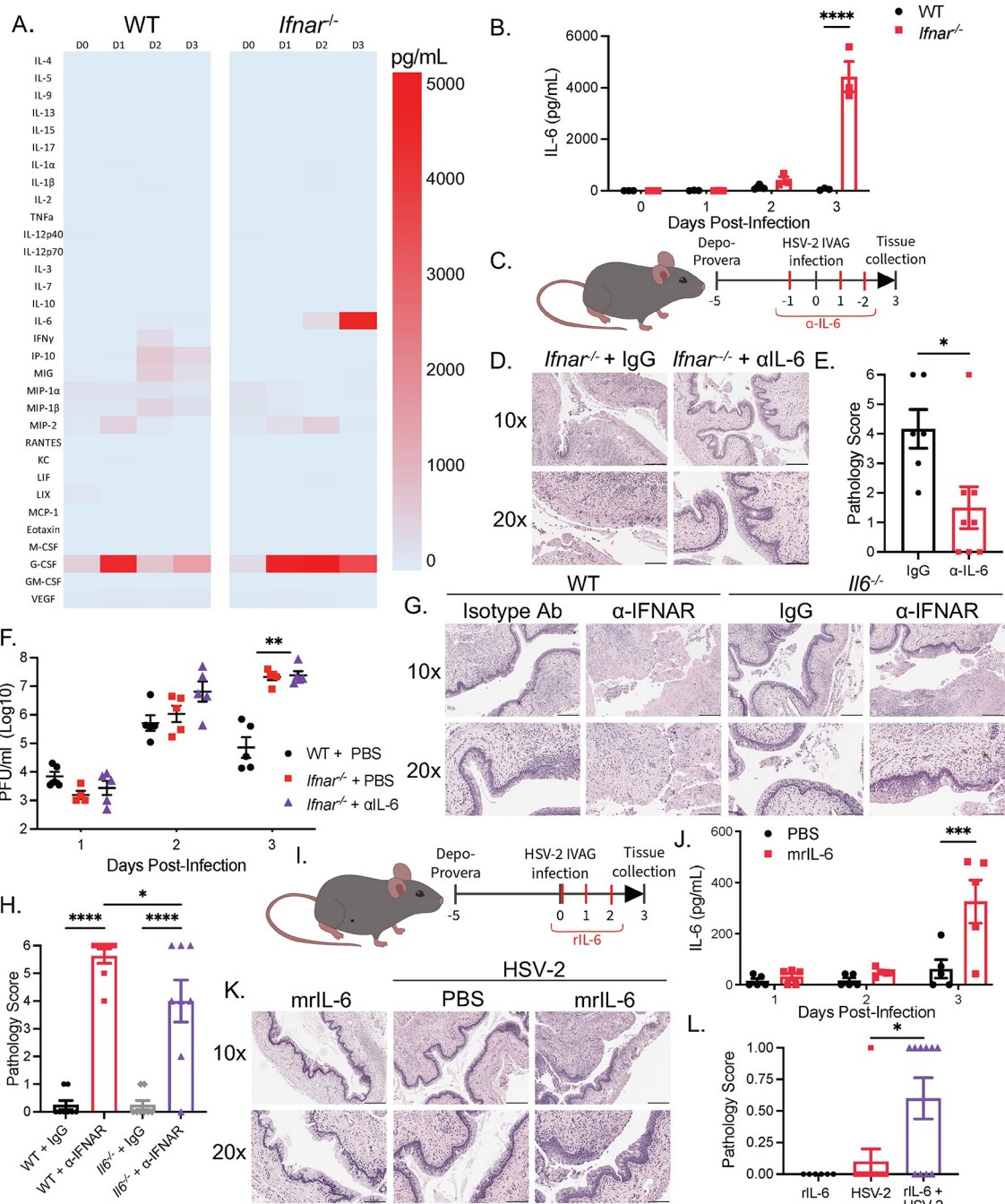

**Fig 4. Type I IFNs regulate IL-6 production to prevent macrophage-mediated immunopathology during viral infection.** (A) Heatmap of cytokine production in vaginal washes following HSV-2 infection. (B) IL-6 cytokine levels in vaginal washes of WT and *Ifnar*−/− mice following HSV-2 infection at 0 to 3 dpi (n = 3). (C) *Ifnar*−/− mice were administered control IgG Ab or α-IL-6 Ab during intravaginal HSV-2 infection. (D and E) H&E staining (D) and pathology score (E) *Ifnar*−/− + IgG, and *Ifnar*−/− + α-IL-6 treated mice at 3 dpi (n = 4–5). (F) Viral titers of WT, *Ifnar*−/− + IgG, and *Ifnar*−/− + α-IL-6 treated mice at 0 to 3 dpi. (G and H) H&E staining (G) and pathology score (H) of WT and *Il6*−/− mice administered isotype IgG or α-IFNAR Ab at 3 dpi. (I) Uninfected and infected WT mice were administered control or mrIL-6 during HSV-2 infection. (J) IL-6 in vaginal washes in PBS and mrIL-6 treated WT mice (n = 5). (K) H&E staining of HSV-2-infected and uninfected WT mice with or without mrIL-6 treatment (n = 5). (L) Pathology score of (K). 10x scale bar represents 200 μm, 20x scale bar represents 100 μm. Data in (B), (E), (F), (H), (J), and (L) are represented as mean ± SEM. $^{*}$p < 0.05, $^{**}$p < 0.01, $^{***}$p < 0.001, and $^{****}$p<0.0001 (B, F, H, J, two-way ANOVA; L, one-way ANOVA; E, two-tailed t-test). See also S4 Fig.

liposomes in NRG mice did not alter IL-6 production, indicating that IL-6 production was not dependent on macrophages, but likely that macrophages respond to IL-6 (S4E Fig). We also observed that neutralizing IL-6 reduced the total number of immune cells and macrophages in *Ifnar*$^{-/-}$ mice without changing the proportion of macrophages at 3 dpi, suggesting a function of IL-6 in promoting leukocyte recruitment (S4F–S4H Fig). IL-6 also has been shown to regulate macrophage function through inducing the expression of the IL-4R and promoting a CD206$^+$ alternative macrophage phenotype [34]. However, while there was an increase in the total number of CD206$^+$ macrophages that was reduced following α-IL-6 administration, the proportion of CD206$^+$ macrophages were unchanged following IL-6 neutralization, suggesting that IL-6 regulated macrophage recruitment above macrophage phenotype (S4I and S4J Fig). Further, macrophages from *Ifnar*$^{-/-}$ mice expressed lower amounts of IL-4R than in WT mice (S4K Fig). Neutralization with α-IL-4Rα Ab also did not alleviate vaginal pathology (S4L Fig). Overall, these results indicate that IL-6 regulates macrophage recruitment and acts independently of type II immune responses to induce immunopathology.

## Type I IFN signaling suppresses MMP proteolytic degradation of tissue following mucosal infection

IL-6 has also been identified to induce macrophage production of MMPs, a class of calcium-dependent zinc-containing endopeptidases that degrade the extracellular matrix (ECM) of the basement membrane [36,37]. During viral infection, MMPs play critical roles in tissue repair, cell migration, regulating cytokine and chemokine activity, and overall leukocyte recruitment during infection [38–40]. Additionally, multiple MMPs, including MMP-2, MMP-9 and MMP-14, have been implicated in tissue pathology in the lung mucosa and female genital tract during infection through mediating inflammation and ECM degradation [41–44]. To investigate the potential role of MMPs in viral-induced pathology, we measured differences in MMP-2 and MMP-9 activity between WT and *Ifnar*$^{-/-}$ mice through a gelatin *in situ* zymography of vaginal tissue cryosections. *Ifnar*$^{-/-}$ mice displayed significantly higher levels of MMP-2 and MMP-9 expressing cells in the submucosa at 2.5 dpi, and incubation with a broad-spectrum metalloproteinase inhibitor 1,10 phenanthroline blocked enzymatic activity (Figs 5A, 5B and S5A). Macrophages were responsible for MMP activity, as clodronate treatment of *Ifnar*$^{-/-}$ mice significantly reduced the proportion of MMP-2 and MMP-9 positive cells in the submucosa (Figs 5C, 5D and S5B). In addition, we determined whether IL-6 was responsible for inducing MMP production from macrophages through the administration of α-IL-6 in *Ifnar*$^{-/-}$ mice. α-IL-6 treated mice showed reduction in MMP-2 and MMP-9 producing cells compared to control mice (Figs 5E, 5F and S5C).

As MMP-2 and MMP-9 activity was upregulated in *Ifnar*$^{-/-}$ mice, and dependent on macrophages and IL-6 signaling, we sought to determine if heightened MMP activity was responsible for virus-induced pathology through administration of doxycycline, known to suppress a broad range of metalloproteinases [45,46]. Doxycycline treatment was able to significantly inhibit MMP-2 and MMP-9 activity in *Ifnar*$^{-/-}$ mice (S5D and S5E Fig). Furthermore, while doxycycline treatment in *Ifnar*$^{-/-}$ mice did not completely abrogate pathology, it resulted in reduced destruction of the vaginal epithelium and the submucosa (Fig 5G and 5H). Doxycycline did not affect viral titers in the vaginal washes, confirming that doxycycline did not exert antiviral effects (Fig 5I). To further confirm the role of MMPs in viral-induced pathology, we administered Ro 28–2653, an MMP inhibitor with preference towards MMP-2, -9, and MMP-14 [47]. Similar to doxycycline, Ro 28–2653 was able to reduce the inflammation and pathology in *Ifnar*$^{-/-}$ mice in response to HSV-2 infection, without affecting viral clearance (Fig 5J–

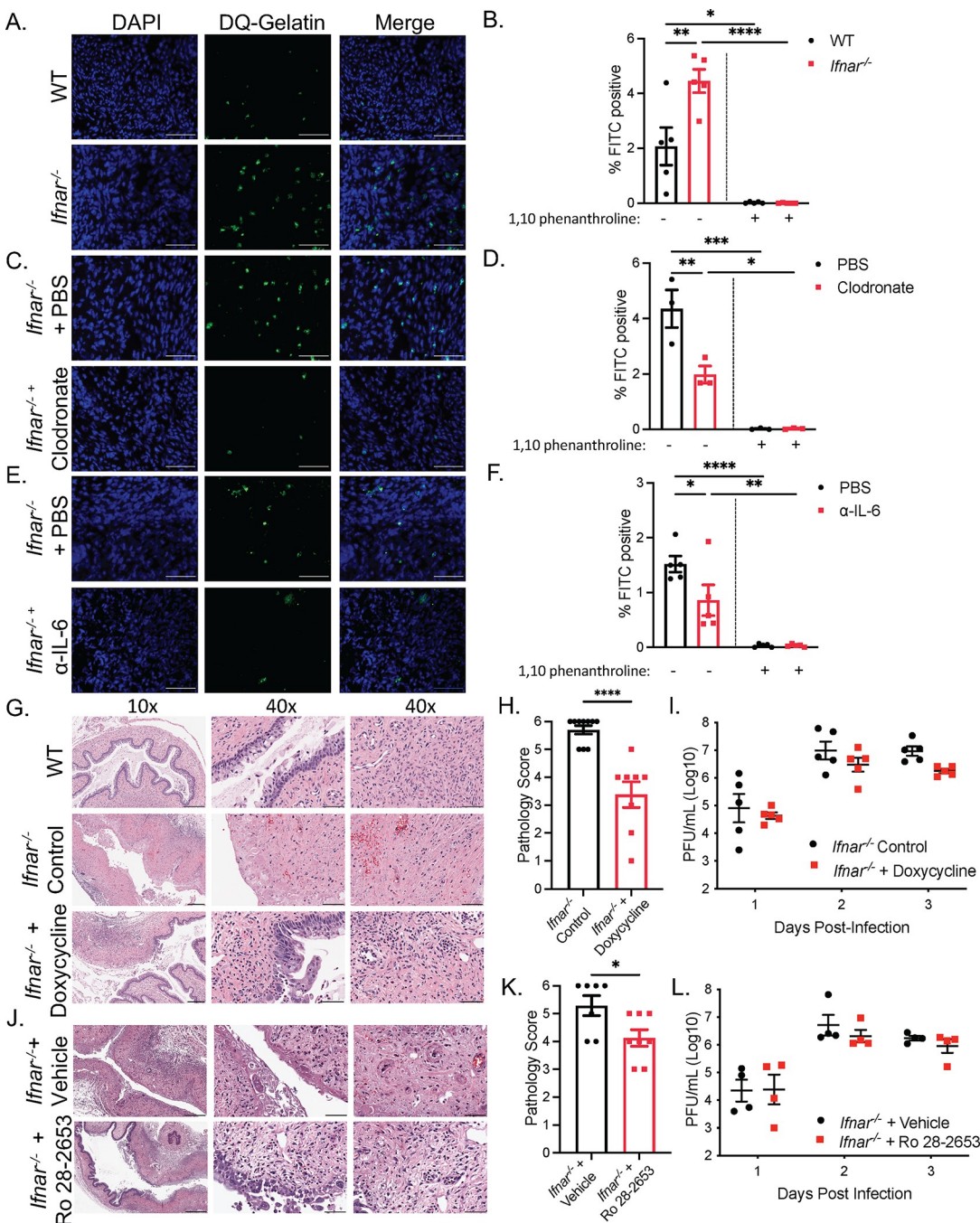

**Fig 5. IL-6 regulates MMP proteolytic degradation of tissue following mucosal infection.** (A) Representative images of *in situ* zymography of gelatinase activity by DQ gelatin (green) colocalizing with DAPI (blue) in the submucosa of HSV-2-infected WT and *Ifnar⁻/⁻* mice at 2.5 dpi. Scale bar represents 50 μm. (B) % FITC positive cells in (A). 3 sections were analyzed and averaged per mouse. 3 separate images per section were quantified and averaged (n = 4). (C) Representative images of *in situ* zymography of HSV-2-infected *Ifnar⁻/⁻* mice administered PBS or clodronate liposomes at 2.5 dpi. (D) % FITC positive cells in (C) (n = 5). (E) Representative images of *in situ* zymography of *Ifnar⁻/⁻* mice administered PBS or α-IL-6 at 2.5 dpi. (F) % FITC positive cells in (E) (n = 3). (G and H) H&E staining (G) and pathology score (H) of vaginal cross-sections of HSV-2-infected WT and *Ifnar⁻/⁻* with or without doxycycline treatment at 3 dpi. Middle: vaginal epithelium, right: vaginal submucosa. 10x scale bar represents 200 μm. 40x scale bar represents 50 μm. (I) Viral titers in vaginal washes of HSV-2-infected WT and *Ifnar⁻/⁻* mice with or without doxycycline treatment from 1 to 3 dpi (n = 5). (J and K) H&E staining (J) and pathology score (K) or vaginal cross-sections of HSV-2-infected *Ifnar⁻/⁻* mice treated with vehicle or Ro 28–2653. (L) Viral titers of vaginal washes of HSV-2-infected *Ifnar⁻/⁻* mice treated with vehicle or RO 28–2653 (n = 4). Data in (B), (D), (F), (H), (I), (K), (L) represented as mean ± SEM. *p < 0.05, **p < 0.01, ***p < 0.001, and ****p <0.0001 (B, D, F, I, L two-way ANOVA; H, K, two-tailed t-test). See also S5 Fig.

5L). These experiments suggest that MMP production by macrophages plays a key role in virus-induced vaginal immunopathology.

## The immunoregulatory function of type I IFNs during viral infection is independent of site of infection

We have shown that type I IFN signaling suppresses IL-6 production to protect against macrophage-mediated tissue pathology during genital HSV-2 infection. To assess whether this phenomenon occurred in a different viral infection at a separate mucosal site, we examined pathology following PR8 IAV infection in *Ifnar*$^{-/-}$ mice and WT mice. We observed significant lung immune-mediated pathology characterized by immune cell infiltration, epithelial necrosis, and hemorrhaging 5 dpi in *Ifnar*$^{-/-}$ mice compared to WT mice (S6A and S6B Fig). In accordance to previous studies [10,11], these were independent of IAV titres as we did not observe differences in viral load between *Ifnar*$^{-/-}$ mice and WT mice (S6C Fig). We then examined whether IL-6 was responsible for lung pathology in *Ifnar*$^{-/-}$ mice. Neutralization of IL-6 in *Ifnar*$^{-/-}$ mice significantly reduced pathology and weight loss without altering viral titers in the lung (Fig 6A–6D). We further assessed whether MMPs were similarly responsibly for lung pathology, and if doxycycline treatment would be equally capable of preventing tissue damage.

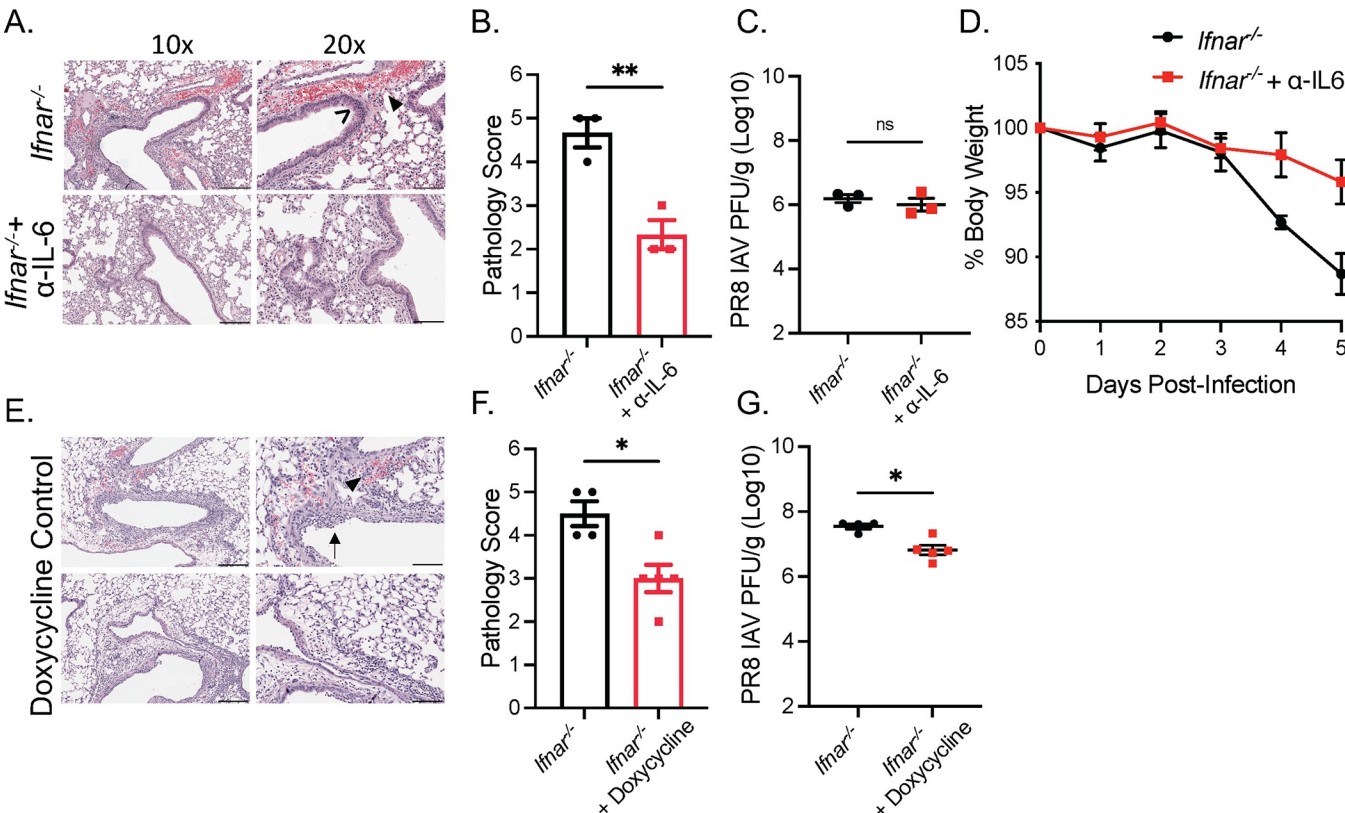

**Fig 6. The immunoregulatory functions of type I IFNs during viral infection is independent of site of infection.** (A) H&E staining of lung cross-sections of *Ifnar*$^{-/-}$ mice administered PBS or α-IL-6 infected with 500 PFU PR8 IAV at 5 dpi (n = 3). (B) Pathology score of (A). (C) Lung viral titers of mice in (A) at 5 dpi. (D) Average body weight of mice in (A) relative to initial starting weight. (E) H&E staining of lung cross-sections of PR8 IAV infected *Ifnar*$^{-/-}$ 5 dpi treated with or without doxycycline in drinking water for -2 to 5 dpi (n = 4–5). (F) Pathology score of (E). (G) Lung viral titers of mice in (E) at 5 dpi (n = 4–5). > indicates necrosis. ► indicates perivascular hemorrhaging. → indicates immune infiltration. 10x scale bar represents 200 μm, 20x scale bar represents 100 μm. Data in (B),(C), (D), (F), and (G) are represented as mean ± SEM. $^*p < 0.05$, and $^{**}p < 0.01$ (D, two-way ANOVA; B, C, F, G, two-tailed t-tests). See also S6 Fig.

*Ifnar^-/-* mice show reduced acute lung injury and mildly reduced viral load following doxycy-cline treatment (Fig 6E–6G). These results suggest that IL-6 and MMP production are capable of tissue pathology in the absence of IFNAR in not only the vaginal mucosa, but also the geo-graphically distant lung mucosa.

## Lungs of severe COVID-19 patients show increased macrophage presence and MMP expression

Severe SARS-CoV-2 infection has been associated with extensive lung inflammation and pneumonia. Type I IFN signaling has also been causally linked to protecting against severe SARS-CoV-2 infection, with severe patients demonstrating a reduced type I IFN signature [7,29,48]. In investigating whether our findings could be extended to severe COVID-19 patients, tissue microarray analysis showed that SARS-CoV-2-induced pneumonia from post-mortem patients coincided with significant infiltration of CD68+ macrophages (S1 Table and Fig 7A and 7B). We further measured MMP mRNA levels in lung tissue samples from post-mortem COVID-19 patients compared to control donors (S2 Table). We observed that COVID-19 patients clustered to express higher levels of MMP-1, -2, -14 and -24, and tissue inhibitors of MMPs (TIMPs) -1 and -2 (Figs 7C and S7, and S3 Table). These results suggest that macrophages and MMPs are strongly associated with COVID-19 pathology and mortality.

## Discussion

The ability to induce and restrict inflammatory immune responses to infection are key to improving survival and preventing immune-mediated pathology to infection. In this study, we describe a novel mechanism where type I IFN signaling regulates macrophage function to inhibit or minimize tissue immunopathology following viral infections in both the vaginal and lung mucosa. In further investigation of how type I IFNs suppress tissue immunopathology, as described in Fig 8, we demonstrate that type I IFNs tightly regulate pathogenic IL-6 inflamma-tory responses in macrophages and their production of MMPs.

The potent antiviral properties of type I IFNs are well studied. As such, the protective role of type I IFN induction to viral infection has been largely attributed to either their antiviral functions or inducing antiviral immune responses [27,49,50]. In contrast, we find that type I IFNs protect against pathogenic responses to viral infection *independent* of their ability to induce antiviral responses. While loss of IFNAR increased viral load in HSV-2 infection, in accordance to previous observations, there were no observable differences in lung viral loads to PR8 IAV infection despite significant pathology only in *Ifnar^-/-* mice [10,11]. Moreover, interventions that suppressed viral-induced pathology to both IAV and HSV-2 infection occurred without impacting viral replication. Both the depletion of macrophages, neutraliza-tion of IL-6, and doxycycline treatment prevented the development of pathology in the absence of improving viral control. These findings suggest that type I IFNs directly protect against pathogenic immune responses that mediate tissue pathology.

Previous studies have attempted to identify the inflammatory processes behind virus-induced pathology. CD8+ and CD4+ T cells as well as ILC2s are previously identified contribu-tors to the development of virus-induced immunopathology in the absence of type I IFN sig-naling [11,51,52]. However, we observed that virus-induced pathology occurred without significant changes in the recruitment of T cells and ILC2s to the vaginal tissue or the presence of any type II cytokines. In sharp contrast to the previously established role of neutrophils in viral-induced pathology [10,53,54], tissue damage occurred independent of neutrophils, as their depletion was unable to suppress virus-induced tissue damage. We have determined that

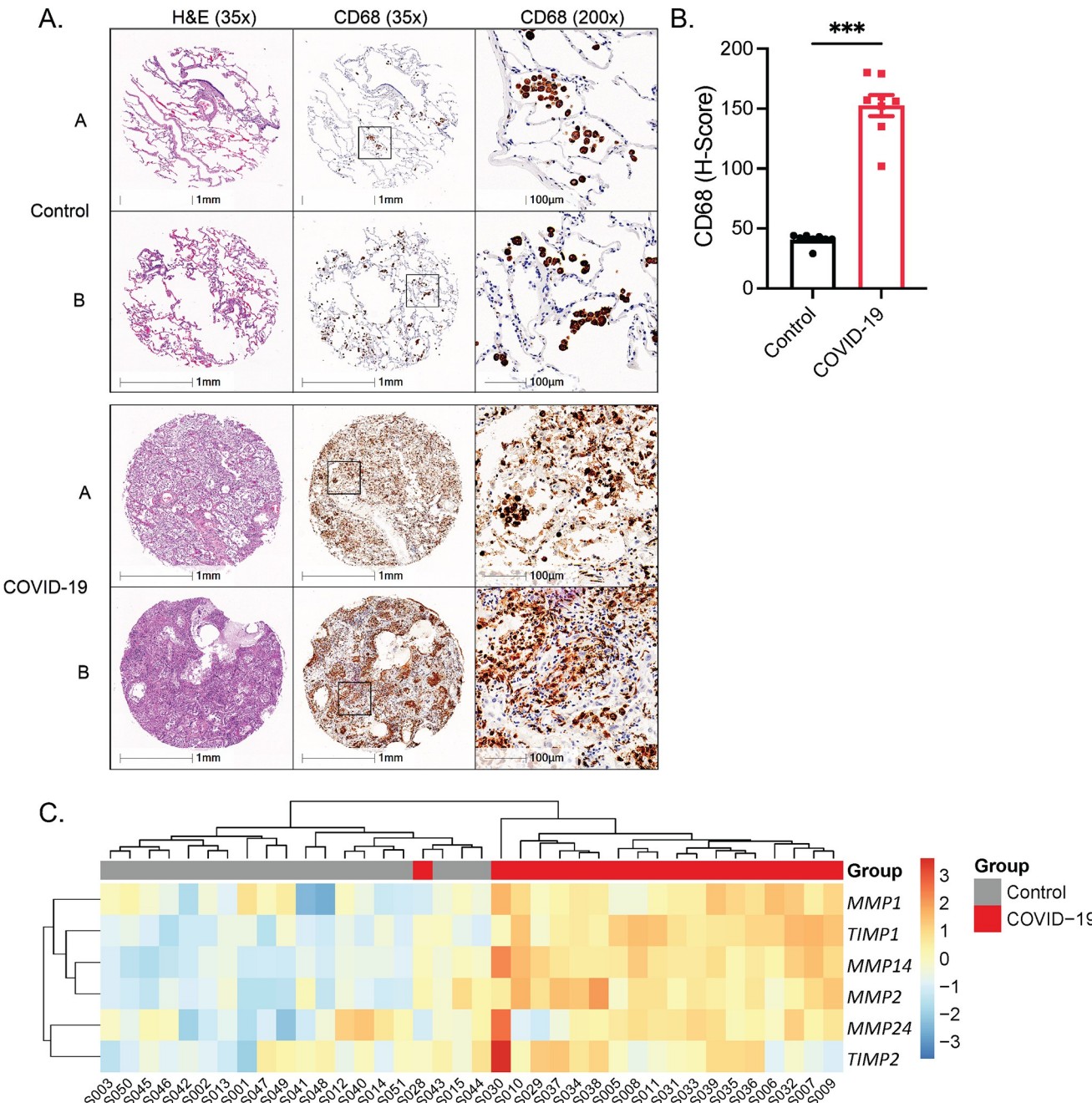

**Fig 7. Severe COVID-19 patients display increased macrophage presence and MMP expression.** (A) Representative images from tissue microarray of lungs from control and COVID-19 patients. A and B represent two separate patients, stained for H&E and CD68. (B) H-Score, quantifying CD68 between control and COVID-19 patients (n = 8). (C) Heatmap analysing significantly upregulated MMP and TIMP gene expression in the lungs between non and COVID-19 patients. Data in (B), represented as mean ± SEM. ***p < 0.001 (two-tailed t-test). See also S7 Fig.

macrophages were the ultimate culprit of pathology to infection, as depletion of macrophages through clodronate liposomes in both the presence and absence of lymphoid cells was able to suppress pathology in the absence of IFNAR.

Accompanying the accumulation of macrophages in the vaginal tissue was a considerable increase in the production of IL-6. IL-6 is often identified as a major mediator of

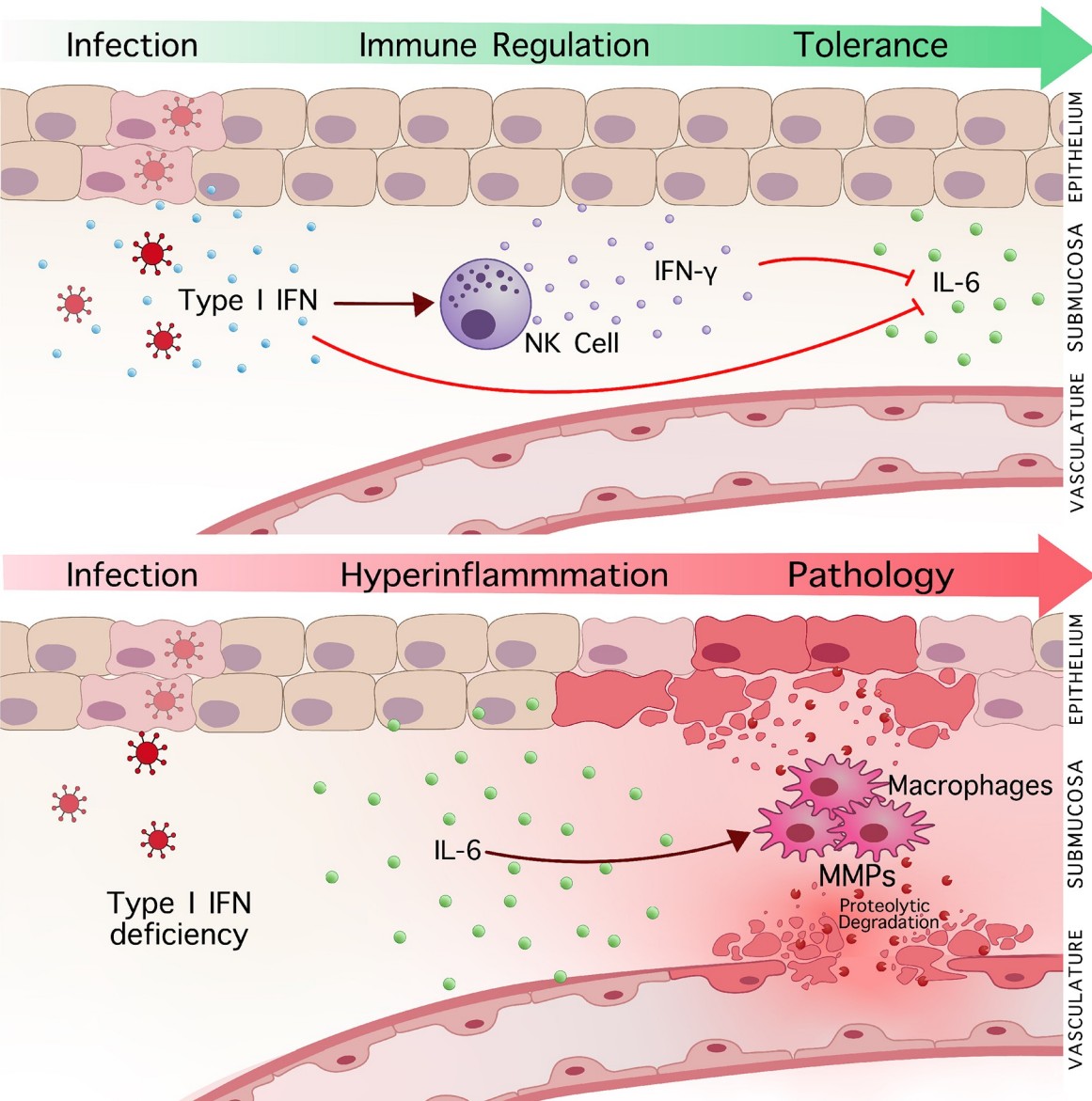

**Fig 8. Type I IFN signaling prevents immunopathology during mucosal viral infection through inhibiting IL-6 and macrophage induced proteolytic degradation.** (Top) Induction of type I IFN signaling suppresses heightened IL-6 production to mucosal viral infection and prevents the development of immune-mediated pathology. Concurrently, induction of NK cell activation and IFN-γ production, facilitated by type I IFN signaling, functions as a secondary mechanism to suppress IL-6-mediated inflammation and tissue pathology to infection. (Bottom) In the absence of type I IFN signaling, heightened IL-6 drives hyperinflammation and virus-induced pathology. Enhanced macrophage recruitment and MMP production by IL-6 induces severe tissue destruction following infection, and inhibition of MMP activity can significantly abrogate virus-induced pathology.

inflammation, and IL-6 production by macrophages has been shown to drive CRS in cancer immunotherapies [25,55–57]. However, we found that macrophages were not the primary source of IL-6, but rather that macrophages responded to IL-6 signaling. IL-6 is likely produced by multiple sources, both hematopoietic and non-hematopoietic, as loss of lymphoid derived immune cells and neutrophils did not abrogate IL-6 driven pathology. The inflammatory effects of IL-6 on immune cells and the development of pathology are poorly defined. In

the context of cancer and chronic inflammatory diseases, IL-6 has been shown to induce the production of MMPs from macrophages [36,37,58]. Meanwhile, IFN-γ antagonizes IL-6 signaling and MMP activity, providing a distinct mechanism through which IFN-γ may act as a redundant inhibitor of IL-6 mediated pathology [59,60].

While MMP activity can be critical for tissue repair following IAV infection [58], excessive proteolytic breakdown of the ECM has been described in virus-induced pathology [40,43]. Previous studies have observed that type I IFNs inhibit TNF-α-dependent MMP-9 activation in tumour cell lines, and IFN therapy can suppress MMP expression in tumour models and during respiratory syncytial virus infection in mice [61–63]. We observed that type I IFNs suppressed IL-6 and macrophage-dependent MMP production following mucosal viral infection. We also observed an increase in macrophages and the transcript expression of MMP-1, -2, -14- and 24, as well as TIMP-1 and -2 in the lungs of human COVID-19 patients. Studies have linked these MMPs to inflammation and pathology following mucosal viral infection. MMP-14 mediated breakdown of the ECM in IAV infection has shown to increase susceptibility to bacterial pneumonia in mice [43]. MMPs have been shown to regulate immune cell recruitment to viral infection, endothelium ECM degradation to facilitate immune cell migration, as well as cleavage of chemokines and cytokines, thereby exacerbating inflammation and immune cell activation [38–40]. Thus, the production of a variety of metalloproteinases following severe viral infection not only weakens the structure of the basement membrane, but would also promote further mechanisms of immune-mediated inflammation.

Therapeutic strategies to prevent hyperinflammation and tissue pathology are needed to effectively treat severe viral infections. However, despite a strong correlation between IL-6 levels and COVID-19 disease severity, current attempts to suppress IL-6 signaling during severe COVID-19 infection have been met with variable results [64–67]. Furthermore, while early type I IFN production seems to correlate with improved COVID-19 disease outcomes, type I IFN therapy is highly dependent on dosage timing in separating their inflammatory versus protective function [7,68]. Inhibition of MMP proteolytic activity offers an alternative and effective therapeutic target downstream of these pathways to suppress immunopathology to infection, without impairing viral clearance. Administration of doxycycline and Ro 28–2653 during HSV-2 and IAV infection substantially reduced the extent of tissue damage in separate mucosal tissues. Increased expression of MMPs in COVID-19 patients also suggests that MMP inhibitors could act as an effective treatment strategy for severe infection in human patients. Currently, the administration of doxycycline in COVID-19 patients has been due to their antiviral properties, and prevention and treatment of secondary bacterial pneumonia [69,70]. Their capacity to control disease severity of both bacterial-induced sepsis and influenza infection has also been attributed to their ability to inhibit mitochondrial protein synthesis [71]. We recognise that as a limitation of the COVID-19 patient data, we cannot conclusively determine that our transcriptional data correlates with increased MMP activity as a cause of pathology due to COVID-19 infection. Our findings support further research into investigating the feasibility and efficacy of MMP inhibitors, such as doxycycline, in the treatment and prevention of viral-induced pathology, and further mechanisms regulating MMP function.

Overall, our findings provide critical insight into the regulation of immune responses to viral infection and identify an immunomodulatory function of type I IFN in the prevention of hyperinflammation and tissue pathology during mucosal viral infection. We present that type I IFNs suppress inflammatory immune responses mediated by macrophages following mucosal infection. These macrophages in the absence of type I IFNs produce MMPs that mediate extensive tissue destruction. Through investigating the immunoregulatory functions of type I IFNs, we have further identified MMP inhibitors as a promising therapeutic strategy to inhibit hyperinflammation and virus-induced tissue pathology. These findings become critical for

developing new therapeutic avenues in a time where emerging viruses are becoming a consistent threat.

## Materials and methods

### Ethics statements

**Animal studies.** All mouse experiments were performed in accordance with Canadian Council on Animal Care guidelines and approved by the Animal Research Ethics Board at McMaster University (AUP 21-04-12). Mice were euthanized by isoflurane followed with cervical dislocation.

**Human studies.** The harvesting and the use of the human lung tissue was approved by the ethics committees of the Hannover Medical School (9621_BO_K_2021). Formal written consent was obtained for the harvesting of all patient samples. For tissue microarray analysis, procedures using human tissues were approved the Hamilton Integrated Research Ethics Board (11–3559 and 13-523-C).

### Mouse model

*Ifnar*$^{-/-}$ mice were bred onto a C57BL/6 background and breeding pairs were provided by Dr. Laurel Lenz (University of Colorado). Colonies of *Ifnar*$^{-/-}$ mice were bred and maintained at McMaster University Central Animal Facility. C57BL/6J (WT) mice were purchased from Charles River Laboratories and Jackson Laboratories. *Il15*$^{-/-}$ mice were purchased from Taconic. *NOD-Rag1*$^{-/-}$*γc*$^{-/-}$ (NRG) and *Il6*$^{-/-}$ mice were purchased from Jackson Laboratory. Mice were housed in specific pathogen-free conditions, a temperature-controlled environment (21 ± 1˚C), on a 12-hour light/12- hour dark cycle, and a maximum 5 mice per cage. Mice were fed an irradiated Teklad global 18% protein diet (cat# 2918) with *ad libitum* access to food and water. Mice that were 6–16 weeks old were infected with either HSV-2 333 or IAV (A/PR8/1934 [H1N1]) virus. All mice were age and sex matched across groups for experiments.

### Human samples

Pulmonary autopsy specimens were collected from patients who died from respiratory failure caused by SARS-CoV-2 infection (n = 19) and compared them to donated age-matched control patients (n = 19) for gene expression analysis. For tissue microarray analysis (TMA) of human lung resected tissue, formalin-fixed paraffin-embedded (FFPE) human lung tissue of patients who died from SARS-CoV-2 (n = 8) were compared to non-involved tissue from lung cancer cases obtained from the biobank for lung diseases from St. Joseph's Hospital in Hamilton, Ontario as controls (n = 8).

### Cell lines and viruses

HSV-2 333 was grown in Vero cells (ATCC) which were maintained in α-MEM supplemented with 1% L-glutamine, 1% penicillin/streptomycin and 5% fetal bovine serum. Infectious viral load was determined by plaque assay and plaque forming units (PFUs) on 12-well plates. Influenza A virus (A/PR8/1934 [H1N1]) was propagated in 10 day old embryonated chicken eggs. Infectious viral load was determined by plaque assay and plaque forming units (PFU) with MDCK cells were grown in MEM supplemented with 1% L-glutamine, 1% penicillin/streptomycin, 1% sodium pyruvate and 10% fetal bovine serum.

### Virus infection and *in vivo* treatments

Female mice were injected subcutaneously (s.c) with 2 mg Depo-Provera 5 days prior to HSV-2 infection. Mice were anesthetized intraperitoneally (i.p) with a mixture of ketamine and

xylazine, and infected with 10 μL $10^4$ PFU HSV-2 333 intravaginally. Vaginal washes were collected through pipetting 30 μl of PBS 5 times in and out of vaginal canal twice for a total of 60 μl. They were collected before infection, and daily following infection. For IAV infection, either female or male mice were anesthetized with isoflurane, and infected intranasally with 20μL in each nostril of varying doses of IAV. To deplete neutrophils, 250 μg of α-Ly6G antibody (BioXCell, 1A8) or isotype control (BioXCell, 2A3) antibody were administered i.p. days -2, -1, and +1 post-infection. To deplete NK cells, 200 μg of α-NK1.1 (BioXCell, PK136) or isotype control (BioXCell, C1.18.4) antibody were administered i.p. days -2, -1, and +1 dpi. To block type I IFN receptor, 500 μg of α-IFNAR (BioXCell, MAR1-5A3) or isotype control (BioXCell, MOPC-2,1) antibody were administered i.p d-1, d0, d1, d2 post-infection. To deplete CD4+ T cells, 200 μg α-CD4 Ab (BioXCell, GK1.5) or isotype control (BioXCell, LTF-2) was administered days -2, -1, and +1 post infection. To block IL-6, 100 μg α-IL-6 (BioXCell, MP5-20F3) or isotype control (BioXCell, HRPN) antibody was administered by i.p day -1, 0, 1, 2 post infection. 1 mg IL-4R (kindly provided by Dr. Manel Jordana, McMaster University) or isotype control (BioXCell, 2A3) was administered 6 days post-infection. To induce IFN-γ production, 1 μg of IL-18 (MBL, B002-5) with 200 ng of IL-12p70 (Peprotech, 210–12) were administered i.p. 1 and 2 dpi. For rIL-6, 1 μg murine IL-6 (Peprotech,216–16) or PBS was administered by i.p day 0 of infection, and twice a day on day 1 and 2 post-infection. To deplete macrophages, 200 μl clodronate or PBS liposomes (Liposoma) were administered i.p. on -2, -1, and +1 dpi, and 15 μl was administered intravaginally -1 dpi. Doxycycline-hyclate (Sigma, D9891) was provided in the drinking water to mice at 60 mg/kg/day from 3 days pre-infection to the end of experiment timeline. Ro 28–2653 (US Biologicals, 464780) reconstituted in 10% DMSO and provided at 60 mg/kg/day by oral gavage from -2 to 2 dpi.

## Viral titration

Viral load of HSV-2-infected mice was detected in the vaginal washes. 5 μL of vaginal lavages were serially diluted from $10^{-2}$ to $10^{-6}$ in 0% α-MEM. Vero cells were seeded on 12-well plate 24 hrs before titration. Media was aspirated, and 200 μL of vaginal lavage dilutions were incubated on Vero cells for 1 hr at 37˚C 5% $CO_2$ with constant rocking, and then overlaid with 2 mL of fully supplemented 5% αMEM. 48 hrs later, cells were fixed and stained with crystal violet, and plaques were quantified. Viral load of influenza-infected mice was detected in lung tissue. Mice were cardiac perfused with 10mL PBS. Lung tissue was homogenised with a Bullet Blender and centrifuged at 5000 rpm for 5 min. Supernatants were titered on confluent MDCKs on 6-well plates. Homogenised tissues were serially diluted from $10^{-1}$ to $10^{-6}$ in infection media (MEM, 0.3% BSA, 0.3% Sodium bicarbonate, 0.001% DEAE-Dextran, 0.25 μg/ml TPCK-Trypsin), and 500 μL of diluted sample was incubated on MDCKs for 1 hr with constant rocking. Media was aspirated and overlaid with 2 mL of 1:1 ratio of 1% agarose and 2X infection media. 48 hrs later, plates were fixed with 4% PFA, the agar overlay was removed, and plaques were visualised with crystal violet.

## Mouse tissue histology

Vaginal canals were isolated 3 dpi and fixed in 2% paraformaldehyde for 48 hrs, and then stored in 70% ethanol. Lung tissue was isolated following cardiac perfusion, and lungs inflated with 10% formalin, fixed 48 hrs, and then stored in 70% ethanol. Tissue was embedded in paraffin, cross-sectioned, and stained with H&E or Picrosirius red (PSR). Slides were scanned with Aperia ScanScope XT (Leica Biosystems). PSR slides scanned at 20x magnification under 45 mm polarized light using VS120 slide scanner microscope (Olympus). PSR positive area was quantified using HALO image Analysis Platform (Indica Labs) and represented as a

percentage of total tissue area, quantified under regular brightfield. Histology images were analysed and taken with Olympus OlyVia. PSR polarised scanned images were false-coloured black and red. Vaginal pathology was scored on a scale of 0 to 6, with separate scores given for epithelium damage and submucosal damage. Epithelial damage was scored from 0 to 3 for no loss, small and localised epithelium loss, 50% epithelium loss, and complete or almost complete loss of epithelium. Submucosal damage was scored from 0 to 3 for no damage, mild damage and hemorrhaging, moderate damage and hemorrhaging, and severe tissue damage and hemorrhaging extending the entire tissue submucosa. Scores were totaled to give the overall pathology score. Lung pathology was scored on a scale of 0 to 5, with individual scores of 0 to 3 given for no, mild, moderate and severe immune cell infiltration and epithelium damage; no, mild, moderate and severe hemorrhaging; and no, mild, moderate, and severe vascular cuffing. Scores were totalled and assigned a score based off of the total: $0 = 0$, $1–2 = 1$, $3–4 = 2$, $5–6 = 3$, $7–8 = 4$ and $9 = 5$.

## Cytokine detection

Vaginal lavages were diluted in PBS and analysed for cytokine levels. Lavages subjected to a 32-plex Luminex (Eve Technologies) including IL-4, IL-5, IL-6, IL-9, IL-13, IFN-γ, TNF-α, and IL-1β. IFN-γ and IL-6 were independently quantified in vaginal lavages using R&D Duo-Set ELISA kits. IFN-α was independently quantified using the R&D ELISA kit.

### *In situ* zymography

Vaginal tissues were isolated 2.5 dpi and snap frozen in OCT (Sakura Tissue-Tek). Vaginal tissue was cut into 7 μM sections and stored in -80˚C. Slides were left for 10 minutes to warm up to room temperature, and OCT was gently rinsed off using PBS. Sections were incubated in 15 μl of 100 μg/mL DQ-gelatin (Enzcheck Invitrogen, D12054) in developing buffer (0.05 M Tris-HCl, 0.15 M NaCl, 5 mM $CaCl_2$, pH 7.4) with 2.5 μg/mL DAPI (Invitrogen) for 4 hrs in humidifying chamber at 37˚C covered from light. Sections were fixed in 4% PFA for 15 minutes and washed 3 x 5 min in PBS. Slides were mounted in Hardset Vectashield. Images were taken at 40x magnification using the Zeiss Axio Imager.M2 and analysed using Fiji ImageJ Software. Images were contrast enhanced for ease of quantification to remove background staining. All images were adjusted the same. For quantification, 3 images per section of the submucosa were taken randomly. The total number of DAPI stained nuclei was used to determine total number of cells in an image. This was determined automatically through applying watershed segmentation on 8-bit B&W images with an automatic threshold applied. FITC+ colocalization with DAPI staining was scored manually. The proportion of FITC+ cells were compiled amongst the three images per section, and then averaged amongst the 3 sections for one sample.

### Vaginal lymphocyte cell isolation

Vaginal tissue was isolated 3 dpi, rinsed from mucus, and processed into small pieces. Tissue was digested in 10 mL of 10% FBS RPMI with 1% L-glutamine, penicillin and streptomycin, HEPES, and 1.3257 mg/mL Collagenase A (Roche) at 37˚C and mechanically stirred for two one-hour incubations. Cells were then passed through a 40 μm filter, spun down, and resuspended for flow cytometry staining.

### Spleen lymphocyte cell isolation

Spleen tissue was gently smashed in culture plate in cold PBS to form single cell suspension. Suspension was filtered and washed with PBS through 70 μM filter. Cells were pelleted, and

red blood cells were lysed with RBC lysis buffer at room temperature for 2–5 minutes. Cells were resuspended with PBS and pelleted again.

## Flow cytometry staining

Live/dead cells were discriminated using with eFluor 780 fixable viability dye (eBioscience,) for 30 mins at room temperature covered from light. Cells were then blocked with anti-CD16/ CD32 (eBioscience) for 20 min. Cells were subsequently stained for extracellular markers anti-mouse CD11b (30-H12), anti-mouse CD45 (30-F11), anti-mouse Ly6G (1A8), anti-mouse CD3 (145-2C11), anti-mouse CD4 (L3T4), anti-mouse CD11c (N418) anti-mouse F4/80 (BM8), anti-mouse lineage cocktail (Biolegend, 133302), anti-mouse Sca-1 (D7), anti-mouse ST1 (D1H9), anti-mouse CD127 (A7R34), anti-mouse 90.2 (30-H12), anti-mouse CD25 (3C7), anti-mouse Siglec-F (S17007L), anti-mouse CD8α (25–0081) for 30 minutes. Samples were fixed for 15 min with 2% PFA. Sample acquisition was conducted using BD LSRFortessa and analysed using FlowJo Software. The IL-4R neutralizing antibody (kindly provided from of Dr. Manel Jordana, McMaster University) was labelled with NHS-Fluorescein (Thermo Scientific).

## Immunoblot analysis of protein

Vaginal tissues were removed at 0 and 1 dpi, snap frozen in liquid nitrogen, and then homogenized. Homogenized tissue was then lysed in 1mL of cold radioimmunoprecipitation assay (RIPA) buffer with protease inhibitors. Protein concentration was then determined using a Bradford Assay. The immunoblot procedure was then conducted as previously described [72]. Briefly, 15–20 μg of protein was loaded on to a 10% SDS-Page Gel and run at 95V for 12 min, and 1 hr for 120V. Gels were transferred onto a nitrocellulose membrane for 1 hour at 400 mA. The membranes were then blocked with LI-COR Odyssey blocking buffer (Mandel). Blots were probed for anti-mouse IL-33 (R&D, AF3626) using a 1:1000 dilution and anti-actin (Santa Cruz Biotechnology, sc-1616) at 1:2000 at 4˚C overnight. Blots were stripped with LI-COR NewBlot Nitro Stripping Buffer. Primary antibodies were detected using an anti-goat IRDye infrared secondary antibody at a 1:10 000 dilution and then imaged with the LI-COR Odyssey infrared scanner. Band densitometry was assessed using software from Image Studio Lite (LI-COR). IL-33 bands were normalised to actin levels, and then quantified as fold changes to d0 vaginal tissue samples.

## Gene expression analysis of COVID-19 pneumonia

RNA was isolate from tissue samples using the Maxwell RNA extraction system (Promega, Madison, Wisconsin). mRNA expression data of pulmonary autopsy specimens was obtained via the nCounter Analysis System (NanoString Technologies, Seattle, WA) using the PanCancer Progression Panel (770 genes including 30 reference genes). Normalization of raw counts was performed using the nSolver analysis software version 3.0 (NanoString Technologies, Seattle, WA) and a modified version of the nCounter advanced analysis module (version 1.1.5). The normalization process included positive normalization (geometric mean), negative normalization (arithmetic mean) and reference normalization (geometric mean) using the 5 most suitable reference genes from the total of 30 available reference genes selected by the geNorm algorithm [73]. Further analysis of the resulting log2 mRNA counts was performed using custom R code. A Shapiro-Wilks test was performed on all intra-group gene expressions which showed that the vast majority of expression data is normally distributed (>85%, $\alpha = 0.05$). Raw counts and normalised data for analyzed genes are found in S3 Table.

## Tissue microarray analysis of pulmonary autopsy specimens

The TMA was imported into HALO and algorithmically segmented into its constituent cores to analyze the TMA on a core-by-core basis. The Multiplex IHC v3.0.4 module was selected to analyze each core to dynamically measure expression of the IHC stain in different compartments of cells. The nuclear stain detection colour was set to (0.959, 0.753, 0.135) corresponding to the RGB colour (11, 63, 221), the IHC stain detection colour (0.268, 0.570, 0.776) corresponding s to the RGB colour (187, 110, 57), and a third stain was added (1.000, 1.000, 1.000) corresponding to the RGB colour (0, 0, 0), which serves as an exclusion stain for regions blackened due to excessive staining. Multiplex IHC v3.0.4 uses the chosen nuclear colour as a basis for locating nuclei during tissue analysis, along with several other parameters to assist with defining the characteristics of the cells. The weight given to the nuclear stain regarding nuclear detection was set to 1, while the weight of the IHC stain was set to 0.45, since many cells possessed IHC stain in the nuclear compartment as well as the cytoplasmic compartment; omitting the latter resulted in a loss of overall cellular detection, especially those with elevated patterns of IHC staining. The nuclear contrast threshold, minimum nuclear optical density, minimum nuclear roundness, and nuclear segmentation were set to 0.515, 0.075, 0.2, and 0.1, respectively. The nuclear size limits were set to no smaller than 5 $\mu m^2$ and no larger than 350 $\mu m^2$, while the option to fill nuclear holes was set to True. Finally, the maximum cytoplasm radius was set to 4 $\mu m$ according to what was representative in the tissue. The IHC stain was given several intensity detection thresholds for H-score analysis. Staining intensities of optical density (OD) 0–0.1499 were categorized as IHC negative cells (or 0), OD 0.15–0.2999 as weakly stained (+1), 0.3–0.4499 as moderately stained (+2), and $\geq 0.45$ as strongly stained (+3), with the condition that the IHC stain must comprise at least 30% of the cell's surface for it to be considered. The algorithm was run across the entire TMA on a core-by-core basis. The TMA is comprised of 2 patient groups: COVID-19 biopsies, control biopsies. Each group was comprised of 8 patients and each patient was comprised of 4 tissues, though some had fewer due to mishaps during the TMA slicing or staining process whereby a tissue is lost or ruined. Each patient had the results from their tissue cores averaged, with each group treating its constituent patients as replicates for statistical analyses.

## Statistical analysis

Statistical differences were analysed using a student's t-test or Mann-Whitney between two groups, a one-way ANOVA for between more than two groups, and a two-way ANOVA for multigroup comparisons. Graphs and statistical analysis was completed using GraphPad Prism 8. Statistical significance is indicated as ns not significant, $^*$p<0.05, $^{**}$p<0.01, $^{***}$p<0.001, $^{****}$p<0.0001.

## Supporting information

**S1 Fig. Type I IFN suppresses vaginal immunopathology during HSV-2 infection.** (A) IFN-α cytokine levels in vaginal washes of WT mice infected with $10^4$ PFU of HSV-2 333 intravaginally (n = 5). (B) IFN-γ cytokine levels in vaginal washes of HSV-2-infected WT, *Ifnar*[-/-], *and Ifnar*[-/-] + mrIL-12/18 mice at 0 to 3 dpi (n = 5). (C) H&E staining of vaginal cross-sections of HSV-2-infected WT, *Ifnar*[-/-] + PBS, *Ifnar*[-/-] + mrIL-12/18, and *Ifnar*[-/-] + mrIL-12/18 + α-NK1.1 at 3 dpi. (D and E) PSR staining of vaginal cross-sections (D) and quantification of % PSR+ to total vaginal area (E) of HSV-2-infected WT, *Ifnar*[-/-], *and Ifnar*[-/-] + mrIL-12/18 at 3 dpi. (F) Viral titers in vaginal washes of HSV-2-infected WT, *Ifnar*[-/-], and *Ifnar*[-/-] + mrIL-12/18 mice at 1 to 3 dpi. H&E scale bar represents 200 μm, PSR scale bar represents 100 μM. Data in (A), (B), (E), (F), are represented as mean ± SEM. $^*$p < 0.05, $^{***}$p < 0.001, and

****p < 0.0001 (A, E, one-way ANOVA; B, F, two-way ANOVA).
(TIF)

**S2 Fig. ILC2 cells and CD4+ T-cells do not mediate the vaginal immunopathology in *Ifnar*-/- mice.** (A) Representative flow plots for CD45+Lin-ST2+CD90.2+ ILC2s in vaginal tissue of HSV-2-infected mice (B to C) Western blot (B) for IL-33 quantification as fold change (C) in vaginal tissue of HSV-2-infected WT and *Ifnar*-/- mice at 0 and 1 dpi. (D) IL-33 cytokine levels in vaginal washes of HSV-2-infected WT and *Ifnar*-/- mice at 0 and 1 dpi. (E) Representative flow plots for CD45+CD3+ T cells gating. (F and G) Representative flow plots (F) and quantification (G) for α-CD4 Ab depletion in blood of HSV-2-infected *Ifnar*-/- mice at 3 dpi. (H) H&E staining of vaginal cross-sections 3 dpi in WT, *Ifnar*-/- + IgG and *Ifnar*-/- + α-CD4 Ab. (I and J) PSR staining (I) of vaginal cross-sections and quantification of % PSR+ to total vaginal tissue area (J) of WT, *Ifnar*-/- + IgG and *Ifnar*-/- + α-CD4 Ab 3dpi. Data in (C), (D), (G), and (J) are represented as mean ± SEM. ***p < 0.001, and ****p < 0.0001 (C, D, two-way ANOVA; H, one-way ANOVA; J, two-tailed t-test).
(TIF)

**S3 Fig. Macrophages mediate pathology, independent of neutrophils and eosinophils in *Ifnar*-/- mice.** (A) Representative flow plots for gating CD45+CD11c-CD11b+Ly6G+ neutrophils in vaginal tissue of HSV-2-infected WT and *Ifnar*-/- mice. (B to C) Total number of neutrophils (B) and proportion of neutrophils to total CD45+ cells (C) in vaginal tissue of HSV-2-infected WT and *Ifnar*-/- mice at 0 to 3 dpi (n = 3). (D to F) Proportion of neutrophils to total CD45+ cells in blood (D; n = 4) spleen (E; n = 4) and vaginal tissue (F; n = 3) in HSV-2-infected WT mice at 3 dpi. (G) H&E staining of vaginal cross-sections of infected *Ifnar*-/- mice with isotype or α-Ly6G Ab at 3 dpi (n = 3). (H and I) PSR staining (H) and % PSR+ to total vaginal tissue area (I) of vaginal cross-sections of HSV-2-infected WT and *Ifnar*-/- mice with isotype or α-Ly6G Ab at 3 dpi. (J) Representative flow plots for gating CD45+CD11b+-Siglec-F+ eosinophils in vaginal tissue of HSV-2-infected WT and *Ifnar*-/- mice. (K to L) Total number of eosinophils (K) and proportion of eosinophils to total CD45+ cells (L) in vaginal tissue of HSV-2-infected WT and *Ifnar*-/- mice at 0, 2 and 3 dpi (n = 3). (M) Representative flow plots for gating CD45+F480+ macrophages in vaginal tissue of HSV-2-infected WT and *Ifnar*-/- mice. (N) Representative flow plots for CD45+F480+ macrophages in vaginal tissue of HSV-2 *Ifnar*-/- mice administered PBS or clodronate liposomes at 3 dpi. Data in (B)–(F), (I), (K), (L) are represented as mean ± SEM. *p < 0.05, and ***p < 0.001 (B, C, K, L two-way ANOVA; D-F, I, one-way ANOVA).
(TIF)

**S4 Fig. IL-6 regulates macrophage-mediated pathology independent of type II immunity.** (A to D) IL-5 (A), IL-13 (B), IL-4 (C), and IL-9 (D) cytokine levels in vaginal washes of HSV-2-infected WT and *Ifnar*-/- mice at 0 to 3 dpi (n = 3). (E) Levels of IL-6 in vaginal washes of HSV-2-infected NRG α-IFNAR Ab with PBS or clodronate liposomes (n = 4). (F) Total number of CD45+ cells in the vaginal tissue of WT and *Ifnar*-/- mice administered PBS or α-IL-6 at 3 dpi (n = 4–5). (G—H) Total number of CD45+F480+ cells (G) and % CD45+F480+ to total CD45+ cells (H) in vaginal tissue of infected WT and *Ifnar*-/- mice administered PBS or α-IL-6 at 3 dpi (n = 4–5). (I-J) Total number CD45+F480+CD206+ cells (I) and % CD206+ to total CD45+F480+ cells (J) in vaginal tissue of infected WT and *Ifnar*-/- mice administered PBS or α-IL-6 at 3 dpi (n = 4–5). (K) IL-4R MFI of vaginal tissue CD45+F480+ cells of HSV-2-infected WT and *Ifnar*-/- mice at 3 dpi (n = 5). (L) H&E staining of vaginal cross-sections for HSV-2-infected WT and *Ifnar*-/- mice administered isotype of α-IL-4R at 3 dpi, scale bar represents 200 μM (n = 5). Data in (A-K) are represented as mean ± SEM. *p < 0.05, **p < 0.01,

***p < 0.001, and ****p < 0.0001 (A-E, two-way ANOVA; F-J, one-way ANOVA; K, two-tailed t-test).
(TIF)

**S5 Fig. Metalloproteinase activity is increased in *Ifnar*<sup>-/-</sup> mice.** (A) Representative control images of in situ zymography submucosa with 1,10 phenanthroline of HSV-2-infected WT and *Ifnar*<sup>-/-</sup> mice at 2.5 dpi. (B) Representative control images of in situ zymography 1,10 phenanthroline of HSV-2-infected *Ifnar*<sup>-/-</sup> mice administered PBS or clodronate liposomes at 2.5 dpi. (C) Representative control images of in situ zymography 1,10 phenanthroline of HSV-2-infected *Ifnar*<sup>-/-</sup> mice administered PBS or α-IL-6 water at 2.5 dpi. (D) Representative images of in situ zymography submucosa of *Ifnar*<sup>-/-</sup> mice administered control or doxycycline in drinking water at 2.5 dpi with or without 1,10 phenanthroline control. (E) % FITC positive cells in (D) (n = 3). Data in (E) represented as mean ± SEM. **p < 0.01, ***p < 0.001, and ****p < 0.0001 (two-tailed t-test).
(TIF)

**S6 Fig. Absence of IFNAR following IAV infection increases lung pathology.** (A) H&E staining of lung cross-sections from WT and *Ifnar*<sup>-/-</sup> mice infected with 300 PFU IAV at 5 dpi. (n = 4). (B) Pathology score of (A). (C) Lung viral titers of mice in (A) at 5 dpi. Data in (B), (C) are represented as mean ± SEM. **p < 0.01 (two-tailed t-test).
(TIF)

**S7 Fig. Increased MMP and TIMP expression in lungs of COVID-19 patients.** (A-K) Box plot of Nanostring log2 counts between control and COVID-19 patients for (A) *MMP14* (B) *MMP1* (C) *MMP3* (D) *MMP17* (E) *TIMP1* (F) TIMP2 (G) *MMP9* (H) IL-6 (I) *MMP2* (J) *MMP24* (K) *TIMP4*.
(TIF)

**S1 Table. Characteristics of COVID-19 patients in TMA.**
(DOCX)

**S2 Table. Characteristics of control and COVID-19 patients in gene expression analysis.**
(DOCX)

**S3 Table. Raw and normalized MMP/TIMP Nanostring data of control and COVID-19 patients.**
(XLSX)

## Author Contributions

**Conceptualization:** Amanda J. Lee, Emily Feng, Ali A. Ashkar.

**Formal analysis:** Amanda J. Lee, Emily Feng, Marianne V. Chew, Danny D. Jonigk, Maximilian Ackermann.

**Funding acquisition:** Ali A. Ashkar.

**Investigation:** Amanda J. Lee, Emily Feng, Elizabeth Balint, Sophie M. Poznanski, Elizabeth Giles, Ali Zhang, Art Marzok, Spencer D. Revill, Fatemeh Vahedi, Anisha Dubey, Ehab Ayaub, Rodrigo Jimenez-Saiz, Joshua J. C. McGrath, Tyrah M. Ritchie, Danny D. Jonigk, Maximilian Ackermann.

**Methodology:** Amanda J. Lee, Emily Feng.

**Resources:** Manel Jordana, Danny D. Jonigk, Maximilian Ackermann, Kjetil Ask, Matthew Miller, Carl D. Richards.

**Supervision:** Manel Jordana, Ali A. Ashkar.

**Visualization:** Emily Feng.

**Writing – original draft:** Amanda J. Lee, Emily Feng, Ali A. Ashkar.

**Writing – review & editing:** Matthew Miller, Carl D. Richards, Ali A. Ashkar.

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
