## [Decision Letter · Decision Letter 0]

30 Nov 2021

Dear Dr. Ashkar,

Thank you very much for submitting your manuscript "Type I interferon regulates proteolysis by macrophages to prevent immunopathology following viral infection" for consideration at PLOS Pathogens. As with all papers reviewed by the journal, your manuscript was reviewed by members of the editorial board and by several independent reviewers. In light of the reviews (below this email), we would like to invite the resubmission of a significantly-revised version that takes into account the reviewers' comments.

We cannot make any decision about publication until we have seen the revised manuscript and your response to the reviewers' comments. Your revised manuscript is also likely to be sent to reviewers for further evaluation.

Sincerely,

Meike Dittmann, Ph.D.

Associate Editor

PLOS Pathogens

Volker Thiel

Section Editor

PLOS Pathogens

Kasturi Haldar

Editor-in-Chief

PLOS Pathogens

orcid.org/0000-0001-5065-158X

Michael Malim

Editor-in-Chief

PLOS Pathogens

orcid.org/0000-0002-7699-2064

Reviewer's Responses to Questions

**Part I - Summary**

Reviewer #1: Amanda Lee et al. demonstrated the key role of type 1 interferons in mucosal viral infections. The absence of type I IFN signaling is linked with several genital tissue destruction during HSV-2 infection. They demonstrated interesting findings in IFNAR-deficient mice linked with immunopathology. However, there are some clarifications needed related to the choice of inhibitors and the role of MMPs and active MMPs in virus infection:

Reviewer #2: The manuscript submitted for review by Lee et al. shows how the absence of IFNAR is associated with immunopathology following viral infection. They use examples of overly active immune responses brought by fatal Ebola infection (discussed in the introduction), and experiments with HSV-2, SARS-CoV2 and Influenza A virus are presented. They discuss how early type I IFN is associated with disease tolerance by SARS-CoV2 whereas delayed or reduced type I IFN is associated with cytokine storms and severe disease. The impactful findings in this study are linking IL6 expression with retarded type I IFN to macrophage MMP expression. This is significant since out of control IL6 expression is associated with hemorrhagic viral disease and ARDs. IL6 has been thrust to the forefront during the COVID19 pandemic because it is seen by the community as one of the principle culprits in ARDS due to SARS-CoV2. I cannot find a paper like this one though there are papers here and there that tackle this area in a nebulous way. However no one has connected the mechanistic dots like this paper, that is, high IFN-I, viral control and low IL6 and macs-MMPs low path, but if low IFN-I leads to high IL6, high macs and MMPS and tissue destruction and mayhem.

In this substantial and compelling paper they use a number of different viruses and models. They start by exploring the intravaginal model of herpes simplex virus type -2 infection. They found that HSV-2 induced pathology was significantly worse in the reproductive mucosae of IFNAR KO female mice. They also used an anti-IFNAR neutralizing antibody in WT mice and showed the same extensive pathology due to HSV-2 infection. There was an increase in CD45+ cells in the mucosae of IFNAR KO mice and a significant decrease in collagen. Other experiments suggested that viral load was not responsible for the increased pathology.

Experiments in the IFNAR KO mice using IL12 and 18 to induce IFNgamma expression reduced HSV-2 mucosae pathology. Pathology was recapitulated with α-NK1.1 mAb treatment of IFNAR DKO mice, that reduced NK cells. Nor were ILC2 cells and Th2 cells involved in the immunopathology, using NRG mouse treated with the IFNAR monoclonal that induced the HSV-2 pathology in NRG mice. They also showed that neutrophils and eosinophils were not involved in the effect that led them to macrophages and somehow, MMPs in influenza mouse model and COVID infected samples.

Assessment and major criticism. The authors show that type I interferon keeps MMP secretion by macrophages at bay. Otherwise there is extensive tissue damage by influx of macrophages secreting gelatinases and other MMPs. The work is well done, with extensive controls, and the manuscript is well written and easy to follow. The data are interesting.

The link between IL6 expression and macrophages during viral infection is likely the most significant of the many discoveries presented by this study. The last section of the paper is where the authors pivot to MMPs but there is no reasoning as to why focus on these proteases. This comes down to improving/providing a lead up rationale for studying MMPs. Superficially, one might argue they used rather blunt tools to inhibit MMPs which I think is the biggest drawback to the paper if it can be considered a drawback. They used 1,10-Phenanthroline and doxycycline both of which have other minor side effects. For example, 1,10-Phenanthroline is a Zn chelator and plucks the Zn out of MMP active sites and possibly other Zn dependent mechanisms. However I don’t see any of these impacting their experimental conclusions owing to the numerous controls. I should add that the authors’ use of these MMP inhibitors was well controlled (doxycycline didn’t affect viral load but reduced epithelial mucosae integrity) and these inhibitors are established in the field with well characterized profiles. The problem with MMPs is that their active sites share very high identity and thus it is very difficult-near impossible to develop an MMP specific inhibitor. With even the most specific MMP inhibitors there is significant cross over inhibition to other MMPs.

The authors used what might be considered by the MMP neophyte as rather blunt instruments to study the involvement of MMPs. The case could be made for further experiments using the various MMP KO mice that are available but this would show them what they already know. It might help identify an MMP(s) responsible for HSV-2 pathology though many papers have been published trying to make a case for one MMP responsible for a given disease but these papers are rarely convincing and usually incorrect. Tissue destruction is due to the expression of a range of MMPs. Therefore I don’t think that further use of MMP KO mice would provide much more insight or change the conclusions of the manuscript. The final figure that looked at MMP expression in COVID patients opens up the topic nicely for further studies to look at the individual MMPs and their substrates in viral infections by others.

Reviewer #3: The manuscript „Type I interferon regulates proteolysis by macrophages to prevent immunopathology following viral infection” by Lee et al., describes prevention of immunopathology after mucosal virus infections by type I IFN. Using IFNAR knockout mice and two infection models, vaginal HSV-2 infection and intranasal infection with influenza virus, they show an IFNAR-dependent control of virus-induced pathology. Knockout of IFNAR results in immunopathology in local tissues after infection which can be prevented by depletion of macrophages and blocking of IL-6 signaling and also by substitution with IFNG. Additionally, they show that the observed immunopathology/tissue damage is promoted by metalloproteinases.

Although the finding that IFNs do regulate immunopathology and metalloproteinases are not new, this comprehensive analysis of the role of type I interferon in immunopathology using two virus infection models in the mouse is a novel and interesting approach. It brings together immunopathology observed after IFN-knockout and/or infection, a correlation of high IL6 levels and immunopathology, and abundant metalloproteinase expression associated with immunopathology. A major weakness of the studies is, that the authors claim or disclaim connections of these pathways without presenting convincing data confirming or disconfirming the connections.

General comment: Although it is nowadays highly fashionable to connect any finding in the lung or in virus infections to SARS-CoV-2 infections, here, it does not strengthen the findings of this paper, especially as the connection is completely based on similarities in phenotypes and not on functional correlates. The SARS-CoV-2 connection regarding for example the treatment with doxycycline should be restricted to the discussion and the data removed.

**Part II – Major Issues: Key Experiments Required for Acceptance**

Reviewer #1: 1- Line 254, 1,10 phenanthroline can inhibit MMPs but it is not a specific inhibitor. It is a broad spectrum and it should be changed to a “broad-spectrum metalloprotease inhibitor.” 1,10 phenanthroline acts via the removal and chelation of the metal ion required for catalytic activity. Therefore, it also inhibits other proteases that degrade the ECM and tissue remodelling such as the ADAMs and ADAMTSs metalloproteases. It can also inhibit bacterial and viral metalloproteases. Therefore, no conclusions to specific inhibition of MMPs can be drawn.

2- Line256, what are gelatinase-positive cells? Do they authors refer as MMP2 and MMP9 producing cells? If it is the case, it is a term that is no longer used and should be referred as MMP2 and MMP9. Also, Gelatin is denatured type IV collagen and is not naturally occurring in vivo and is an artifact of gel denaturation or frozen histology as performed by in situ zymography.

3- Doxycycline has been demonstrated to inhibit MMP activity but again, it is not a specific MMP inhibitor. It can inhibit multiple other proteases including bacterial collagenases, among many others. Therefore, the conclusion (Line 263-264) that MMP production by macrophages plays a role in virus-induced vagnial immunopathology is not supported by the data. A more accurate statement would be that “metalloproteases” (from mice, bacteria or virus) can regulate this process. The same issue with Line 280-281. It should be changed to metalloproteases and not MMPs specifically.

4- Figure 7, the authors show data that mRNA levels of MMP1, -2, -14 and -24 are elevated. An elevation of mRNA levels does not imply increased MMP activity. Also, TIMP1 and TIMP2 are also elevated. TIMPs are inhibitors of MMP activity. Therefore, there is no evidence on line 294-295 that the MMP activity is elevated in post-mortem COVID19 patients. To assess MMP activity, one would need to use activity-based probes that bind active MMPs. This conclusion cannot be supported by the data presented here. Same apply for Line 369-370. There is no evidence by these data that doxycycline would inhibit MMPs activity alone, without side effects, in addition that the authors did not demonstrate increased MMP activity but a change in mRNA levels.

5- MMPs do many more functions in virus infection other than ECM degradation. Examples include: 1- Marchant et al. (2014) Nature Medicine, 20(5) 493-502; 2- Cheung C, et al (2008) Circulation 117:1574–1582; 3- Yang S, et al. (2019) J Hepatol 71:685–698; 4- Zhang K, et al. (2003) Nature Neuroscience 6:1064–1071. 5- Vergote D, et al. (2006) Proc Natl Acad Sci 103:19182–19187. Therefore, the statement that MMPs degrade the matrix, ECM, and can result in tissue destruction is outdated. There are multiple other roles by MMPs during viral infection such as regualtion and processing of chemokines, cytokines, cell surface receptors, etc.

6- Multiple MMPs can be produced by other immune cells such as neutrophils (MMP8, MMP9 and MT6-MMP), monoyctes (mulitple MMPs), T cells (mulitple mmps), etc. It is true that there are increased recruitement of macrophages but how do the authors know that this effect is entirely linked to macrophages and not partially due by neutrophils, monocytes or T cells?

7- Abstract and overall conclusion, Line 43-44: the authors state that “uncover MMPs as therapeutic targets towards viral infections” is not supported by the inhibition data they demonstrated or the post-mortem COVID19 patients analysis.

8- Post-mortem information of COVID19 should be presented. What were their age and sex?

Reviewer #2: No further experiments

Reviewer #3: Major points:

1) Induction of IFNG abolishes virus-induced immunopathology in IFNAR-knockout mice. The authors’ point that IFNA and IFNG-dependent signaling are independent, but redundant pathways suppressing virus-induced pathology is not convincingly shown. It is known that IFNG can upregulate IL-6 and that I can affect macrophage activity. IFNG is a known inducer of metalloproteinases. The authors prresent no convincing experiment which excludes that the IFNA-dependent protection from immunopathology is not due to downregulation of IFNG.

2) Figure 5: The authors claim that Figure 5 shows that type I IFN signaling suppresses IL-6 mediated MMP degradation of tissue. The figure only shows that macrophages mediate this degradation without showing for example the effect of anti-IL6 antibodies as used in figure 4. This has to be added to make their point.

3) In their second model, intranasal IAV infection, they claim to see the same effects in IFNAR knockout mice as they saw in vaginal HSV2 infection. Here, they only show IL6- dependency of MMP-dependent tissue damage. To make their point both in Figures 4 and 5, clodronate and anti-IL6 inhibition have to be shown.

**Part III – Minor Issues: Editorial and Data Presentation Modifications**

Reviewer #1: Minor comments:

1-Line 40 and line 48 (and throughout the manuscript): As per Uniprot nomenclature, MMPs abbreviation should be changed to matrix metalloproteinases (MMPs)

2- Figure 5G: Doxycycline has a capital “O” that should be changed.

3- S5 fig, Line 531: the word “specific” should be removed as 1,10 phenanthroline is not a specific MMP inhibitor.

4- S7 Fig and Fig7, Line 544-544: Gene names should be italicized.

Reviewer #2: Abstract. The second sentence is awkward and not entirely true. IFN type I can make one more susceptible to Lupus for example. I would change the wording from, “less is known about how IFNs prevent immunopathology and increase disease tolerance,” to something like “…less is known about how IFNs affect immunopathology and related disease.”

Introduction. I suggest being more focused in the introduction of the paper. There is discussion of influenza, Ebola and SARS-CoV2 but no introduction to HSV-2, a very different virus in its own right. One may mention these other viruses but I strongly suggest focusing the introduction on HSV-2 since this is the research topic.

Line 120. They need a lead up in this paragraph of the results section to explain what IL15 does and why they looked at IL15 DKO mice. This will make the paper easier to read as the rationale at this point in the paper is not clear.

Line 247. There is no lead up explanation as to why the authors chose to look at MMPs so this needs to be corrected in a revision. There are other proteases secreted by macrophages so why MMPs? This is of course rhetorical in that MMPs are some of the best characterized proteases in immunity. Though it is not for me to write the paper but the authors can provide their own reasoning in a lead up paragraph. I suggest citing the relevant literature as there are a few papers already looking at mechanisms of MMPs during viral infection.

The figures are neat and compelling but the font is a touch small and fuzzy in a lot of the panels. I suggest increasing the font size in these figures to make the figures easier to read. The HandE and other tissue stains are also fuzzy and impossible to interpret. The authors have done themselves no favors (or the reviewer) in submitting figures like this which I find quite aggravating. The figures all need to be fixed in this regard.

Materials and methods. This section appears to be the most lacking in detail. I don’t believe there is a size limit for PLoS Pathogens so the authors can put significantly more work into this section rather than merely citing previous papers.

Reviewer #3: Minor points:

1) Using IL15 knockout mice to proof that elevated titers in HSV2 are not inducing immune pathology is not really suitable for showing that the immunopathology is not driven by elevated titers. IL15 knockout mice show differences in innate immunity, which are not defined here. Showing the IFNAR knockout plus clodronate (Figure 3 F) should be sufficient to make this point.

2) Figure 4: Why should anti-IFNGR antibodies affect pathology in IL6 knockout mice if IL6 is a major downstream player in IFGNR knockout mice immunopathology? As to be expected, the effects were minor.

3) Discussion: In the discussion the authors claim that type I IFN-dependent inhibition of MMPs is a novel finding. This is not true. There are numerous publications e.g. Sanceau et al., 2002 in Mechanisms of Signal Transduction that type I IFN does inhibit metalloproteinases. Surprisingly, none of them was cited by the authors. Selected citations should be added and discussed.

4) In contrast to the description in the text, Fig S1F Fig 1L show identical data sets.

5) Many figure numbers in the text are mixed up and are additionally not systematically organized (e.g. Fig. 3I).

6) Using a mouse virus instead of only human viruses adapted to mouse would increase the quality of the data and should at least be discussed.

7) A figure depicting their model of the IFN-IL6-macrophage-metalloproteinase network would have been helpful.

PLOS authors have the option to publish the peer review history of their article (what does this mean?). If published, this will include your full peer review and any attached files.

Reviewer #1: No

Reviewer #2: **Yes: **David James Marchant

Reviewer #3: No
---

## [Decision Letter · Decision Letter 1]

24 Mar 2022

Dear Dr. Ashkar,

We are pleased to inform you that your manuscript 'Type I interferon regulates proteolysis by macrophages to prevent immunopathology following viral infection' has been provisionally accepted for publication in PLOS Pathogens.

Best regards,

Meike Dittmann, Ph.D.

Associate Editor

PLOS Pathogens

Volker Thiel

Section Editor

PLOS Pathogens

Kasturi Haldar

Editor-in-Chief

PLOS Pathogens

orcid.org/0000-0001-5065-158X

Michael Malim

Editor-in-Chief

PLOS Pathogens

orcid.org/0000-0002-7699-2064

Reviewer Comments (if any, and for reference):

Reviewer's Responses to Questions

**Part I - Summary**

Reviewer #1: We thank the reviewers for addressing our suggestions which has led to a much improved manuscript.

Reviewer #2: The manuscript submitted for review by Lee et al. shows how the absence of IFNAR is associated with immunopathology following viral infection. When the host has high type-I IFN, there is viral control and low IL6 and low macrophage secreted MMP levels resulting in less pathology, but if there is low type I IFN or response this leads to high IL6, high macrophage tissue infiltration causing tissue destruction through MMP expression and activity.

Reviewer #3: The authors adequately adressed all issues, I critized in the review of the original submission and as far as I can judge also addressed all the comments of the other reviewers. The mansucript has been highly improved and many points are now dicussed in a more differentiated way.

**Part II – Major Issues: Key Experiments Required for Acceptance**

Reviewer #1: We thank the reviewers for addressing our suggestions which has led to a much improved manuscript.

Reviewer #2: I have no further experiments to suggest for acceptance.

Reviewer #3: (No Response)

**Part III – Minor Issues: Editorial and Data Presentation Modifications**

Reviewer #1: all corrected.

Reviewer #2: I have no minor issues with this paper

Reviewer #3: I would like to insist to remove the word "novel" from the sentence line 340 of the discussion.

PLOS authors have the option to publish the peer review history of their article (what does this mean?). If published, this will include your full peer review and any attached files.

Reviewer #1: **Yes: **Antoine Dufour

Reviewer #2: **Yes: **DAVID JAMES MARCHANT

Reviewer #3: No

---

## [Editor Report · Acceptance letter]

27 Apr 2022

Dear Dr. Ashkar,

We are delighted to inform you that your manuscript, "Type I interferon regulates proteolysis by macrophages to prevent immunopathology following viral infection," has been formally accepted for publication in PLOS Pathogens.

Best regards,

Kasturi Haldar

Editor-in-Chief

PLOS Pathogens

orcid.org/0000-0001-5065-158X

Michael Malim

Editor-in-Chief

PLOS Pathogens

orcid.org/0000-0002-7699-2064